# Factored Gossip DiLoCo: Reducing Blocking Communication in DiLoCo

**Chamin Hewa Koneputugodage** [1]  **Thalaiyasingam Ajanthan** [1]  **Sameera Ramasinghe** [1]
**Hadi Mohaghegh Dolatabadi** [1]  **Shamane Siriwardhana** [1]  **Gil Avraham** [1]  **Violetta Shevchenko** [1]  **Karol Pajak** [1]
**James Snewin** [1]  **Alexander Long** [1]

## Abstract

To make large-scale distributed training practical outside high-bandwidth datacenters, we must reduce blocking, high-volume synchronization. While DiLoCo communicates infrequently, its outer synchronization remains bandwidth-heavy and brittle to stragglers and transient failures. We relax exact synchronization to *approximate synchronization* via mixing/gossip, which degrades gracefully under delays and communication failures. This allows us to factorize DiLoCo synchronization into a *non-blocking* mixing step that overlaps computation with no staleness, and a *blocking* mixing step that tightens worker agreement, yielding a tunable trade-off between compute utilization and optimization stability. On up to billion-parameter language models in low-bandwidth settings, our framework substantially improves compute utilization compared to DiLoCo, with training progress ranging from comparable to closely matching it, and is more robust to failures.

## 1. Introduction

Scaling large language model (LLM) training continues to unlock major gains in capability (Grattafiori et al., 2024; Liu et al., 2024a; Brown et al., 2020), but centralized clusters face hard communication, energy, and infrastructure bottlenecks. This motivates distributed training, as well as decentralized training across geographically and administratively separated collaborators (Ryabinin et al., 2020; 2023; Avraham et al., 2025). To be practical, such settings must mitigate communication bottlenecks, especially in low-bandwidth, consumer-grade settings.

One popular approach is DiLoCo (Douillard et al., 2023), which reduces communication frequency by running many

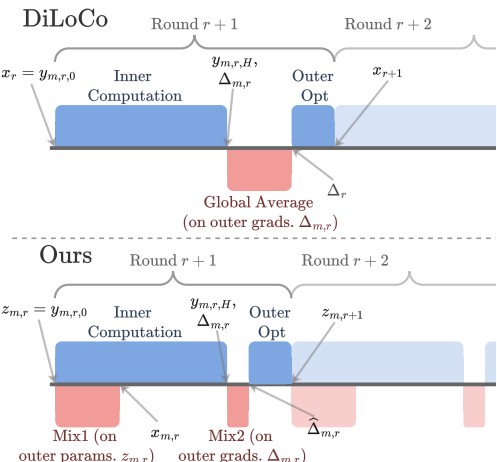

*Figure 1.* **Comparison of DiLoCo and our approach.** We replace exact per-round synchronization with approximate synchronization via two operations: Mix1 mixes previous outer-step parameters (non-blocking, overlaps communication without temporal staleness) and Mix2 mixes the latest outer gradient (blocking). This enables us to use overlapped global averaging for Mix1 and a minimal amount of mixing that is sufficient for stability for Mix2.

local optimization steps between synchronization steps. However, its outer synchronization remains (i) high-volume and blocking (it stalls computation), and (ii) brittle: bandwidth-efficient global collectives (e.g., ring all-reduce (Patarasuk & Yuan, 2009)) are global barriers, dominated by the slowest link, and can abort on a single link failure, requiring retries (Meta Platforms, 2024). Because DiLoCo relies on successful synchronizations to reset disagreement, such failures cannot be ignored without risking instability.

In this paper, we address these problems with *approximate synchronization*. DiLoCo already tolerates some disagreement between workers during local steps, and its global synchronization resets this to zero. We relax exact global synchronization to non-global synchronization that keeps disagreement low. Since synchronization is fundamentally an averaging operation, mixing/gossip is the natural relaxation (Lian et al., 2017; Koloskova et al., 2020). Furthermore, it is less sensitive to delays and degrades gracefully under transient failures, where missed exchanges simply yield weaker mixing rather than aborting the round.

---

[1]Pluralis Research. Correspondence to: Chamin Hewa Koneputugodage <chamin@pluralis.ai>.

*Proceedings of the 43$^{rd}$ International Conference on Machine Learning*, Seoul, South Korea. PMLR 306, 2026. Copyright 2026 by the author(s).

Tolerating approximate synchronization also lets us factorize outer synchronization into two components: a *non-blocking* mixing step (Mix1) that overlaps with computation without temporal staleness, and a blocking mixing step (Mix2) that more strongly enforces consensus for stability. We can choose the mixing operator for each step; in particular, we use overlapped global averaging for Mix1 to bound disagreement with minimal compute impact, and use blocking non-global mixing in Mix2 to tighten this bound and improve stability, at the cost of lower compute utilization.

Our experiments show large compute-utilization gains with convergence ranging from comparable to closely matching DiLoCo, improved robustness to communication failures, and an empirical study of how disagreement bounds relate to optimization stability and the trade-off between compute utilization and stability.

## 2. Related Work

**Synchronous Data Parallel.** Synchronous data parallel (sync-DP) averages gradients globally before each update (Sergeev & Balso, 2018; Goyal et al., 2017). This is equivalent to single-worker optimization with a larger effective batch, but requires expensive global communication.

**Local SGD and Periodic Model Synchronization.** Methods that reduce synchronization *frequency* by taking multiple local steps have a long lineage: early parallel-SGD work periodically averaged worker parameters (Zinkevich et al., 2010; McDonald et al., 2010), DOWNPOUR added asynchronous parameter-server SGD (Dean et al., 2012), Elastic Averaging SGD (EASGD) (Zhang et al., 2015) replaced direct averaging with an elastic pull toward a slowly-moving center variable, and blockwise model-update filtering (BMUF) (Chen & Huo, 2016) added a global momentum term over averaged updates that presaged the outer-optimizer view. Federated Learning (FL) (Li et al., 2021; Kairouz et al., 2021) later refined these ideas: Local SGD (Stich, 2019) and FedAvg (McMahan et al., 2017) use periodic global parameter averaging, while FedOpt (Reddi et al., 2021) treats the multi-step parameter change as an *outer gradient* for a global outer optimizer. DiLoCo (Douillard et al., 2023) adapts FedOpt to LLM training with AdamW (Loshchilov & Hutter, 2019) as the inner optimizer and Nesterov accelerated SGD (Nesterov, 1983; Kallusky et al., 2025) as the outer optimizer, enabling many more inner steps between global synchronizations.

**Decentralized SGD and Gossip-based Optimization.** Gossip-based methods replace global synchronization with sparse peer-to-peer communication, improving consensus via local mixing over a graph rather than enforcing exact global agreement each round (Kempe et al., 2003; Blot et al., 2016; Lian et al., 2017; Assran et al., 2019; Koloskova et al., 2020). Several works combine gossip with periodic averaging (Wang & Joshi, 2021; Koloskova et al., 2020; Chen et al., 2021; Guo et al., 2022), but target the FL objective rather than reducing blocking communication. The closest to our approach is NoLoCo (Kolehmainen et al., 2025), which has a similar factorized structure: they apply Mix2 with pairwise gossip, use a damped correction in the outer update that incorporates Mix1 with pairwise gossip, and using pipeline randomization to reduce worker divergence. Unlike our work, they do not study varying Mix1/Mix2 or explicitly optimize for reduced blocking and higher compute utilization.

**Overlapping Communication and Computation.** Overlapping communication and computation masks communication overhead and improves utilization, especially for high-latency cross-site links. To preserve semantics, communication is launched as early as possible but awaited at required synchronization points (e.g., gradient bucketing in sync-DP, or overlapping pipeline-parallel transfers with compute). In practice, these synchronization points limit overlap. Asynchronous methods instead increase overlap by tolerating temporal staleness, e.g., parameter-server SGD (Recht et al., 2011; Agarwal & Duchi, 2011; Dean et al., 2012; Paine et al., 2013; Zhang et al., 2013; Feyzmahdavian et al., 2015; Mishchenko et al., 2022), asynchronous gossip (Lian et al., 2018), and asynchronous pipeline scheduling (Huang et al., 2019; Narayanan et al., 2019; 2021; Ajanthan et al., 2025a), often with delay correction (Zheng et al., 2017; Ajanthan et al., 2025a).

**Overlapping Communication and Computation with DiLoCo.** Asynchrony has also been explored for DiLoCo-style methods. Liu et al. (2024b) push outer gradients to a server and mitigate outer-momentum staleness via less frequent momentum updates and fewer local steps for slow workers. Ajanthan et al. (2025b) improve upon this by adding look-ahead delay correction that extrapolates in the negative momentum direction. Streaming DiLoCo (Douillard et al., 2025) overlaps outer-gradient averaging with a few local steps, then merges the resulting stale parameters with the latest via a weighted average. Kale et al. (2025) increases overlap to a full outer step and reduces staleness by updating the averaged outer gradient with each worker's latest contribution. To our knowledge, our approach is the first to overlap communication and computation in a DiLoCo-based setting without introducing temporal staleness.

In this paper, we focus on *exact communication* (no quantization or sparsification) to isolate the effects of topology and synchronization structure. Compression is largely orthogonal and could be added on top. We also do not study staleness-based/asynchronous update schemes, which target a different axis by tolerating stale information rather than restructuring synchronization.

## 3. Preliminaries

**DiLoCo.** Following Khaled et al. (2025), let $x_r \in \mathbb{R}^d$ be the globally synchronized outer parameters at round $r$, and $y_{m,r,h}$ the inner parameters on worker $m \in \{1, ..., M\}$ after $h \in \{0, ..., H\}$ local steps, using stochastic input batch $b_{m,r,h}$. DiLoCo updates on each worker $m$ as

$$y_{m,r,0} = x_r \tag{1}$$

for $h = 0, 1, \ldots, H - 1$ in sequence:

$$g_{m,r,h} = \nabla_{y_{m,r,h}} \mathcal{L}(y_{m,r,h}, b_{m,r,h}) \tag{2}$$

$$y_{m,r,h+1} = \text{InnerOpt}\,(y_{m,r,h}, g_{m,r,h}) \tag{3}$$

$$\Delta_{m,r} = y_{m,r,H} - y_{m,r,0} \tag{4}$$

$$\Delta_r = \frac{1}{M} \sum_{m=1}^{M} \Delta_{m,r} \tag{5}$$

$$x_{r+1} = \text{OuterOpt}\,(x_r, -\Delta_r)\,. \tag{6}$$

**Gossip-based Mixing.** Given values $v_{m,r}$ on each worker $m$ at round $r$ (e.g., parameters or gradients), gossip-based mixing has each worker compute

$$v_{m,r+1} = \text{Mix}_{W_r}(v_{m,r}) := \sum_{n=1}^{M} w_{m,n,r} v_{n,r} \tag{7}$$

where $W_r \in \mathbb{R}^{M \times M}$ has entries $w_{m,n,r}$ and is symmetric and doubly stochastic. When $W_r$ is implicit, we write $\text{Mix}_r$. Global averaging corresponds to $W_r = W = \frac{1}{M} \mathbf{1}_M \mathbf{1}_M^T$, while local/gossip averaging uses sparse $W_r$. Let $X_r = [x_{1,r}, ..., x_{M,r}] \in \mathbb{R}^{d \times M}$ and $\overline{X}_r = [\overline{x}_r, ..., \overline{x}_r] = X_r \frac{1}{M} \mathbf{1}_M \mathbf{1}_M^\top$, where $\overline{x}_r = \frac{1}{M} \sum_{m=1}^{M} x_{m,r}$. After one mixing step the stacked values are $X_r W$, ideally equal to $\overline{X}_r$.

Mixing quality is summarized by the contraction factor $0 \le \rho \le 1$, the reduction in squared distance to the global average. For mixing matrices $W \sim \mathcal{W}$, define the expected contraction factor $\overline{\rho}$ by

$$\mathbb{E}_W \left[ \|X_r W - \overline{X}\|_F^2 \right] \le \overline{\rho} \|X_r - \overline{X}_r\|_F^2. \tag{8}$$

Thus, global averaging corresponds to $\rho = 0$, and no averaging corresponds to $\rho = 1$. Apart from these two, we also use synchronous pairwise random gossip: each communication round samples a random matching (set of disjoint pairs), and within each pair workers average their values. This has expected contraction factor $\overline{\rho}_{\text{pair}} = \frac{M-2}{2(M-1)} \approx \frac{1}{2}$ (Boyd et al., 2006), so the high-probability upper bound on the contraction factor is likely to satisfy $\frac{1}{2} < \rho_{\text{pair}}^{\uparrow} < 1$.

**Gossip-based Optimization.** The classic way to do gossip averaging is to mix the parameters after each optimizer step (Jin et al., 2016; Blot et al., 2016; Koloskova et al., 2020):

$$g_{m,r} = \nabla_{x_{m,r}} \mathcal{L}(x_{m,r}, b_{m,r}) \tag{9}$$

$$z_{m,r+1} = \text{Opt}\,(x_{m,r}, g_{m,r}) \tag{10}$$

$$x_{m,r+1} = \text{Mix}_r\,(z_{m,r+1})\,. \tag{11}$$

Another variation is to average parameters before each optimization step (Jiang et al., 2017; Lian et al., 2017)

$$x_{m,r} = \text{Mix}_r(z_{m,r}) \tag{12}$$

$$g_{m,r} = \nabla_{z_{m,r}} \mathcal{L}(z_{m,r}, b_{m,r}) \tag{13}$$

$$z_{m,r+1} = \text{Opt}\,(x_{m,r}, g_{m,r}) \tag{14}$$

where $x_{m,0} = z_{m,0} = x_0$. Unrolled, the order of operations $(g_{m,r}, z_{m,r+1}, x_{m,r+1})$ is still the same, However, now the gradient $g_{m,r}$ is calculated at $z_{m,r}$, while being applied in the optimizer step to $x_{m,r}$. The benefit of this approach is that it allows the mixing communication and the gradient computation to overlap. However, in the classic formulation the gradient in the $r + 1^{\text{th}}$ round has information from other workers after their optimization step in round $r$, while in this variation the gradient does not. Thus, while the gradient is not temporally stale, it is noisier than in the classic version.

## 4. Factored Gossip DiLoCo

We first discuss our factored gossip formulation in Section 4.1. We then show that it allows for a more communication friendly (but noisier) rearrangement of DiLoCo in Section 4.2, and further discuss implications on overlapping communication in Section 4.3. Finally, we discuss the importance of consensus, as well as how to control and measure its effects, in Section 4.4.

### 4.1. Our Factored Gossip Formulation

The mixing operation in Equation (11) can be considered as a combination of the mixing operation in Equation (12) and mixing on the computed gradients plus resulting update. Thus, we consider a factorization into Mix1, a non-blocking parameter mixing operation like Equation (12), and Mix2, a blocking gradient mixing operation:

$$x_{m,r} = \text{Mix1}_r\,(z_{m,r}) \tag{15}$$

$$g_{m,r} = \nabla_{z_{m,r}} \mathcal{L}(z_{m,r}, b_{m,r}) \tag{16}$$

$$\Delta_{m,r} = \text{Opt}(z_{m,r}, g_{m,r}) - z_{m,r} \tag{17}$$

$$\widehat{\Delta_{m,r}} = \text{Mix2}_r\,(\Delta_{m,r}) \tag{18}$$

$$z_{m,r+1} = x_{m,r} + \widehat{\Delta_{m,r}}. \tag{19}$$

Here, $z_{m,r}$ represents the result after the $r^{\text{th}}$ update step, and $x_{m,r}$ represents a further mixed representation of it. As the gradient in the $r + 1^{\text{th}}$ optimization round is calculated at $z_{m,r}$, the communication for $x_{m,r}$ can overlap that computation. Thus, the $r + 1^{\text{th}}$ optimization round has two mixing operation: Mix1 which is non-blocking communication of parameters after the previous update step, and Mix2 which is blocking communication of parameter differences (essentially an outer gradient) after the latest update step.

Applying this for our DiLoCo case gives

$$x_{m,r} = \text{Mix1}_r\left(z_{m,r}\right) \tag{20}$$

$$y_{m,r,0} = z_{m,r} \tag{21}$$

for $h = 0, 1, \ldots, H - 1$ in sequence:

$$g_{m,r,h} = \nabla_{y_{m,r,h}} \mathcal{L}(y_{m,r,h}, b_{m,r,h}) \tag{22}$$

$$y_{m,r,h+1} = \text{InnerOpt}\left(y_{m,r,h}, g_{m,r,h}\right) \tag{23}$$

$$\Delta_{m,r} = y_{m,r,H} - y_{m,r,0} \tag{24}$$

$$\widehat{\Delta_{m,r}} = \text{Mix2}_r\left(\Delta_{m,r}\right) \tag{25}$$

$$z_{m,r+1} = \text{OuterOpt}\left(x_{m,r}, -\widehat{\Delta_{m,r}}\right). \tag{26}$$

Vanilla DiLoCo corresponds to not having Mix1 and Mix2 being a global average. Since all workers start from the same initialization, a global Mix2 enforces exact consensus and makes Mix1 redundant, though at the cost of heavy blocking communication. In contrast, Mix1 can overlap DiLoCo's large number of inner steps $H$, thus it has a lot of time before it can block the next outer step.

This factorization raises natural questions: *How necessary is blocking Mix2 for convergence? Can it be removed if Mix1 is sufficiently global (or perhaps fully global)? When, if ever, is it beneficial to use both mixing operations?*

For intuition, in Section E we analyze a simplified setting where both optimizers are SGD, under strong stability assumptions. Let $\rho_1, \rho_2 \in [0, 1]$ denote the contraction factors of Mix1 and Mix2. For $\rho_1 < 1$, we show that for any $r, h$ the expected average distance between any two workers is bounded:

$$\mathbb{E}\left[\frac{1}{M^2} \sum_{m,s=1}^{M} \|y_{m,r,h} - y_{s,r,h}\|^2\right] \le 2\eta^2\sigma^2 H F, \tag{27}$$

where $F = 1 + \frac{\rho_2}{1-\rho_1}$, $\eta$ is the inner learning rate, and $\sigma^2$ bounds gradient variance. We also show that the vanilla DiLoCo case ($\rho_1 = 1, \rho_2 = 0$) lies on the boundary of this regime: the global Mix2 enforces exact consensus at the start of every outer round, so the bound reduces to $2\eta^2\sigma^2 H$ (i.e. $F = 1$ by convention), matching Khaled et al. (2025).

The bound highlights that both Mix1 and Mix2 reduce consensus error, but Mix1 is critical: if $\rho_1 \to 1$ without a global Mix2, the consensus error can diverge. In contrast, $\rho_2$ mainly tightens the bound, setting $\rho_2 = 1$ keeps it finite but looser. Thus we consider the following four main configurations

- **DiLoCo**: Our framework with $\rho_1 = 1, \rho_2 = 0$, so $F = 1$
- **LocalM1M2**: Our framework with pairwise gossip for both Mix1 and Mix2, so $\rho_1 = \rho_2 = \rho_{\text{pair}}^{\uparrow}$ and thus $F > 2$. Note that this baseline is very similar to NoLoCo (Kolehmainen et al., 2025), though with dampening and pipeline randomization removed.

- **GlobalM1LocalM2**: Our framework with global averaging for Mix1 and pairwise gossip for Mix2, so $\rho_1 = 0$, $\rho_2 = \rho_{\text{pair}}^{\uparrow}$ and $\frac{3}{2} < F < 2$. Note that this has the same amount of blocking communication as LocalM1M2.
- **GlobalM1**: Our framework with global averaging for Mix1 and no Mix2, thus no blocking communication, so $F = 2$. Note that this is a fixed factor regardless of $M$.

We also consider the following variant to demonstrate that the JS-distance metric introduced in Section 4.4 is a useful measure for improving consensus and convergence:

- **GlobalM1LocalM2-subset**: global Mix1 with pairwise-gossip Mix2 applied only to a subset of parameter blocks (token embeddings, LM-Head, and the first 2 transformer layers, 43% of parameters at 1.5B). The subset is selected offline via per-block JS-distance sensitivity (Section D.3).

Note that GlobalM1 is still approximate synchronization, since the parameters are not synchronized at the end of the outer step. Only by having Mix2 as global will the outer step be globally synchronized. However, reducing Mix2 (by decreasing $\rho_2$) reduces blocking communication while linearly decreasing the consensus error in Equation (27). Detailed per-worker pseudo-code, including the asynchronous launch of Mix1, is in Algorithm 1 (Section A).

### 4.2. Rearranged DiLoCo

We now show that our framework with Mix2 removed and Mix1 as global averaging is essentially DiLoCo with rearranged steps and noisier outer gradients. Note that we have globally consistent parameters after Mix1, so $x_{m,r} = x_r$. We also make the assumption that the outer update step is linear in the outer gradient and has global parameters, thus $\text{OuterOpt}(x_r, g_{m,r}) = x_r - \gamma(a_r g_{m,r} + b_r)$ where $a_r$ and $b_r$ do not depend on $m$. Then our formulation is

$$y_{m,r,0} = z_{m,r} \tag{28}$$

$$\Delta_{m,r} = y_{m,r,H} - y_{m,r,0} \tag{29}$$

$$z_{m,r+1} = \text{OuterOpt}\left(x_r, -\Delta_{m,r}\right) \tag{30}$$

$$= x_r - \gamma\left(-a_t\Delta_{m,r} + b_t)\right) \tag{31}$$

$$x_{r+1} = \frac{1}{M}\sum_{m=1}^{M} z_{m,r+1} \tag{32}$$

$$= \text{OuterOpt}\left(x_r, -\Delta_r\right) \tag{33}$$

where $\Delta_r = \frac{1}{M}\sum_{m=1}^{M}\Delta_{m,r}$. This is equivalent to the original DiLoCo algorithm in Equation (6) with two differences. First, the initial point of the inner optimization steps $y_{m,r,0}$ starts at the local value $z_{m,r}$ rather than its globally consistent value $x_r$. This makes the local outer gradients $-\Delta_{m,r}$ more noisy, they are still proper outer gradients but they

are computed starting at noisy point $z_{m,r} = x_r + \epsilon_m$ rather than globally consistent point $x_r$. Second, the global average happens after the optimizer step, rather than before, and is a parameter average not a gradient average. Due to the assumptions on the optimizer step $x_{r+1}$ has the exact same value as DiLoCo, the result of the outer optimizer on $x_r$ and the global average of the local outer gradients, though with noisier local outer gradients. However this rearrangement allows the communication of the global averaging to overlap the next $(r + 1^{\text{th}})$ optimization round's computation.

In the appendix (Section B) we show that DiLoCo's outer optimizer satisfies linearity, and to satisfy global parameters we need to also synchronize the state. However, as discussed there, this greatly increases the communication volume, and its performance does not justify it.

### 4.3. Differentiating Non-Blocking Communication and Staleness

The analysis as rearranged DiLoCo in the previous section provides a clear comparison between our non-blocking operation and asynchronous methods for DiLoCo that introduce staleness. In order for there to not be staleness, the $r^{\text{th}}$ outer optimization step that yields $z_{m,r}$ (Equations (26) and (30)) must come before the computation (local steps) in the $r + 1^{\text{th}}$ round. This is a hard synchronization point. As explained in Section 2, asynchronous methods like Douillard et al. (2025) do not obey this, introducing staleness. We compare against Streaming DiLoCo's staleness-introducing communication–computation overlap approach in Section B.2.

Our formulation factorizes the communication coming up to that point as Mix1 and Mix2. As explained earlier, having Mix1 overlap computation does not introduce staleness because $z_{m,r}$ has the same number of outer update steps as $x_{m,r}$. Thus, workers compute their next outer gradient from the latest *local* update $z_{m,r}$, rather than the latest *global* update $x_{m,r}$, so $z_{m,r}$ is *temporally correct* but only represents a *local* average rather than a globally average. This introduces a starting-point noise term, not a temporal lag, so the outer gradients are noisier than DiLoCo's but never stale. After the outer gradients have been computed and potentially mixed with Mix2, workers then apply them to a more global starting point due to Mix1. Note that Mix2 by definition reduces noise in the outer gradients, so it can directly compensate for the disparity between $x_{m,r}$ and $z_{m,r}$.

### 4.4. Consensus Control with Gossip

Kong et al. (2021) observe that the performance gap between local and global averaging is highly related to the consensus distance between workers, which they define as the average L2 distance between worker parameters and the mean parameters. They theoretically show, and empirically validate, that when consensus distance is below a critical threshold, local averaging converges as fast as global averaging. Khaled et al. (2025) similarly connect consensus distance to optimization performance, where their convergence rates depend on a result similar to Equation (27).

As a result, it is important to understand and measure the consensus distance. During training we log the more efficient L2 consensus distance of Kong et al. (2021)

$$CD_{L2}(X) = \frac{1}{M} \sum_{m=1}^{M} \|x_m - \overline{x}\|_2. \qquad (34)$$

As we show in Section 5, the ordering of our logged L2 distance matches $F$ in Equation (27).

However, the trend in L2 distance does not match the trend in training performance. We designate this to the theory using strong assumptions to relate L2 distance to function value and its loss. In their proof (Khaled et al., 2025), via assumptions on smoothness, convexity and Lipschitz constants, bounds on L2 distance are converted to bounds on gradients and then to bounds on function loss.

Thus, we also measure functional distance via the Jensen–Shannon (JS) distance between each worker's per-token output distribution and that of the averaged parameters,

$$CD_{JS}(X) = \frac{1}{M} \sum_{m=1}^{M} JS\left(f(x_m, b), f(\overline{x}, b)\right) \qquad (35)$$

where $f(x, b) := \text{softmax}(\text{logits}(x, b))$ produces a per-token probability distribution and $b$ is a held-out batch, sampled once and reused at every measurement (full details in Section D). As we show in Section 5 and Figure 6, this tracks training performance much better: instability and loss spikes typically coincide with spikes in JS distance.

The JS-distance metric can also be used to determine that not all parameter blocks contribute equally to functional disagreement. We exploit this with the GlobalM1LocalM2-subset configuration introduced earlier, which applies Mix2 only to the most sensitive blocks. The per-block sensitivity analysis, the choice of subset, and a subset-size ablation are in Section D.3. The configuration's results in Table 1 and Figures 2 to 4 demonstrate that the JS-distance metric is useful in determining a good compromise between reduced communication volume and training performance.

## 5. Results

As we target public, internet-scale decentralized training, our experimental setting differs substantially from DiLoCo (Douillard et al., 2023; 2025). We assume single-GPU workers on consumer networks, roughly 50Mbps–1Gbps, rather than multi-GPU datacenter nodes with 10–100Gbps links. This constrains model size due to

*Table 1.* Summary of our experimental configurations and results. The top section contains the main results comparing our four configurations, with ablations in the following sections. Here, *local* denotes pairwise gossip averaging, *global* denotes full averaging across all replicas, and GBS is the global batch size (per-worker batch is GBS/$M$). The two $M = 16$ blocks correspond to two scaling strategies: GBS=512 keeps the global batch fixed (so the per-worker batch halves relative to $M = 8$), while GBS=1024 keeps the per-worker batch fixed to the $M = 8$ baseline (so the global batch doubles). All runs use AdamW (inner LR $3\times10^{-4}$) as the inner optimizer and SGD with Nesterov momentum (outer LR 0.7, momentum 0.9) as the outer optimizer, matching DiLoCo's hyperparameters. Overall, overlapped global Mix1 communication substantially boosts compute utilization, while adding Mix2 improves consensus and convergence at the cost of utilization (since it is blocking).

| Config (row) | $M$ | $H$ | GBS | Mix1 | Mix2 | Val PPL at $T$ tokens | | Compute Util. at | | Max Distance | |
| | | | | | | $T$=10B | $T$=30B | 100Mbps | 200Mbps | L2 | JS |
|---|---|---|---|---|---|---|---|---|---|---|---|
| **Sync DP** | 8 | - | 512 | - | - | 17.53 | 14.85 | 1% | 1% | 0 | 0 |
| **DiLoCo** | 8 | 100 | 512 | none | global | 18.65 | 15.52 | 36% | 53% | 65.5 | 0.21 |
| **LocalM1M2** | 8 | 100 | 512 | local | local | 19.37 | 15.83 | 49% | 66% | 234.8 | 0.24 |
| **GlobalM1** | 8 | 100 | 512 | global | none | 19.42 | 15.91 | 56% | 100% | 145.1 | 0.34 |
| **GlobalM1LocalM2** | 8 | 100 | 512 | global | local | 18.80 | 15.60 | 36% | 66% | 106.5 | 0.22 |
| **DiLoCo** | 8 | 500 | 512 | none | global | 22.67 | - | 74% | 85% | 148.1 | 0.26 |
| **LocalM1M2** | 8 | 500 | 512 | local | local | 23.13 | - | 83% | 91% | 407.1 | 0.26 |
| **GlobalM1LocalM2** | 8 | 500 | 512 | global | local | 22.78 | - | 83% | 91% | 201.7 | 0.26 |
| **GlobalM1** | 8 | 500 | 512 | global | none | 23.17 | - | 100% | 100% | 285.9 | 0.42 |
| **DiLoCo** | 8 | 1k | 512 | none | global | 28.17 | - | 85% | 92% | 204.2 | 0.27 |
| **LocalM1M2** | 8 | 1k | 512 | local | local | 28.82 | - | 91% | 95% | 435.9 | 0.27 |
| **GlobalM1LocalM2** | 8 | 1k | 512 | global | local | 28.13 | - | 91% | 95% | 283.2 | 0.27 |
| **GlobalM1** | 8 | 1k | 512 | global | none | 28.65 | - | 100% | 100% | 384.8 | 0.49 |
| **Sync DP** | 16 | - | 512 | - | - | 17.22 | - | 0% | 1% | 0 | 0 |
| **DiLoCo** | 16 | 100 | 512 | none | global | 19.99 | - | 21% | 34% | 69.0 | 0.23 |
| **LocalM1M2** | 16 | 100 | 512 | local | local | 22.24 | - | 24% | 49% | 546.3 | 0.26 |
| **GlobalM1** | 16 | 100 | 512 | global | none | 21.39 | - | 26% | 52% | 154.3 | 0.35 |
| **GlobalM1LocalM2** | 16 | 100 | 512 | global | local | 20.70 | - | 17% | 34% | 109.9 | 0.22 |
| **Sync DP** | 16 | - | 1024 | - | - | 18.17 | - | 1% | 1% | 0 | 0 |
| **DiLoCo** | 16 | 100 | 1024 | none | global | 20.91 | - | 34% | 51% | 66.0 | 0.20 |
| **LocalM1M2** | 16 | 100 | 1024 | local | local | 23.01 | - | 49% | 66% | 483.2 | 0.24 |
| **GlobalM1** | 16 | 100 | 1024 | global | none | 22.26 | - | 52% | 100% | 144.2 | 0.31 |
| **GlobalM1LocalM2** | 16 | 100 | 1024 | global | local | 21.48 | - | 34% | 66% | 108.9 | 0.22 |
| **GlobalM1LocalM2-subset** | 8 | 100 | 512 | global | local | 19.15 | - | 45% | 82% | 123.6 | 0.22 |
| **GlobalM1LocalM2-subset** | 8 | 500 | 512 | global | local | 23.06 | - | 92% | 96% | 239.7 | 0.28 |

per-device memory limits, though this could be alleviated using model parallelism. Pipeline parallelism is especially favorable here because bandwidth is amortized per stage, and its longer forward–backward step times hide more communication (Ryabinin et al., 2023; Ramasinghe et al., 2025). However, this is beyond the scope of this paper.

### 5.1. Training experiments

We train a 1.5B parameter Llama 3 model (Grattafiori et al., 2024) on FineWeb (Penedo et al., 2024), and run experiments for 10B tokens, with main experiments replicated to 30B tokens (compute optimal). In our setup, we use 8 workers, each with a 40GB A100 GPU. We train with a sequence length of 1024 and a batch size of 512. We use a peak learning rate of $3 \times 10^{-4}$ and a warmup of 1000 steps, with linear learning rate decay to zero. In order to plot validation loss, since FineWeb does not have a validation set we make a running validation set by randomly excluding some documents from the training set (deterministic across runs).

We show our main results in Table 1 (top section) and Fig-

ure 2. As mentioned in Section 4.1, LocalM1M2 is very similar to NoLoCo, and thus functions as a strong baseline. We also compare against a reimplementation of NoLoCo, though without pipeline randomization as it is orthogonal to our DP focus, in Section B.1.

Table 1 shows a clear utilization-performance trade-off under these low bandwidth settings. Sync-DP achieves the best validation perplexity (17.53) but essentially no utilization ( 1%). DiLoCo (H=100) is the main baseline, reaching moderate utilization (36% @100Mbps, 53% @200Mbps) with reasonable perplexity (18.65/15.52 at 10B/30B tokens). Purely local mixing (LocalM1M2) improves utilization (49%/66%) but has worse perplexity (19.37/15.83) and much larger divergence. Using Mix1 alone (GlobalM1) maximizes utilization (56% @100Mbps, 100% @200Mbps) but suffers a slight penalty in perplexity (19.42/15.91) and increases divergence. Adding Mix2 restores stability: GlobalM1LocalM2 improves perplexity to 18.80/15.60 while keeping DiLoCo-like utilization at 100Mbps and improving it at 200Mbps. The subset-Mix2 variant further boosts utilization (45%/82%) with a modest perplexity cost.

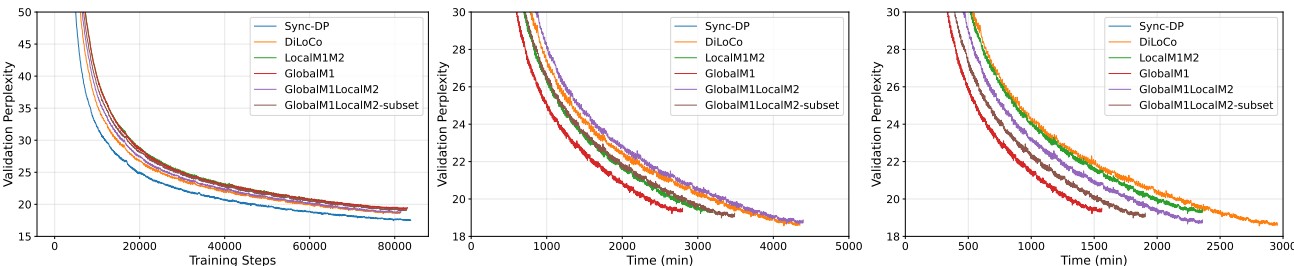

*Figure 2.* **Validation perplexity curves**. **(Left)** Validation perplexity per step. **(Middle)** Validation perplexity over time with 100Mbps bandwidth. **(Right)** Validation perplexity over time with 200Mbps bandwidth. While GlobalM1LocalM2 has the smallest perplexity gap to DiLoCo, removing blocking communication yields much faster convergence with respect to wall time.

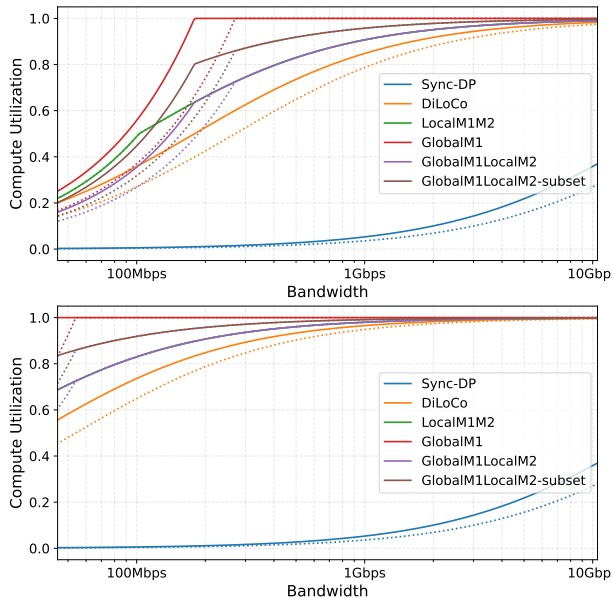

*Figure 3.* Compute utilization under different bandwidths for $M = 8$ workers and either **Top** $H = 100$, or **Bottom** $H = 500$ inner steps. Dotted lines show results with 5% communication failure rate. When Mix1 is fully hidden, utilization approaches 100%, and under failures the drop is smaller than all-reduce baselines until communication exceeds the available overlap.

## 5.2. Compute Utilization

Following [Douillard et al. (2025)](#), we build a simple compute-utilization simulator and report on a range of bandwidth. Our simulator has four components, forward/backward compute, outer optimization compute, Mix1 communication, and Mix2 communication, and assumes a fixed per-link bandwidth between any pair of workers. We additionally provide a simulation of a 5% communication failure rate.

When Mix1's global all-reduce is fully hidden under the local-step compute, our methods achieve substantially higher utilization than DiLoCo. In our settings this occurs for $H = 100$ once bandwidth exceeds roughly 200Mbps, and for $H = 500$ across the full bandwidth range. More pre-

cisely, at our 1.5B-parameter $M = 8$ setup the minimum $H$ for full Mix1 overlap is $H \approx 180$ at 100Mbps and $H \approx 90$ at 200Mbps. In this regime, GlobalM1 reaches 100% utilization and adding various levels of Mix2 (in regards to number of parameters) reduces utilization smoothly. However, when Mix1 cannot be fully overlapped due to extremely low bandwidth, utilization can drop sharply, and can even fall below DiLoCo, since we now pay for two communication phases. Notably, a bandwidth-optimal all-reduce requires at most about twice the bandwidth of pairwise gossip, so the utilization differences are driven primarily by *blocking* and restart behavior rather than raw bandwidth alone. Translated to wall-clock time, training to 1T tokens at 200Mbps with $M = 8$ and $H = 100$ takes approximately 16 weeks for GlobalM1 versus 29 weeks for DiLoCo.

Under simulated failures, all-reduce must restart while mixing can proceed with effectively weaker mixing, making DiLoCo's utilization degrade across bandwidth. In contrast, our methods only lose utilization once failures push communication time beyond what computation can hide, and LocalM1M2 is unaffected since it avoids all-reduce entirely.

**Memory overhead.** Our methods add no GPU-memory overhead beyond DiLoCo: the outer optimizer state lives on CPU and is communicated via GLOO, so the only extra per-step allocation is a single parameter-sized buffer for the outer update. In our setup, peak VRAM during local steps is $28.29$ GiB and rises to $29.89$ GiB during the outer step, identical across DiLoCo and our four configurations.

## 5.3. Distance Metrics

We plot L2 and JS consensus distances in Figure [4](#). Sync-DP has zero distance by construction, and DiLoCo returns to zero at the end of each outer step ($H = 100$). Both metrics spike at the end of warmup and settle into a steady-state level. For DiLoCo, distances quickly rise to steady state as workers drift during local computation. The non-global blocking methods show the same qualitative behavior in L2, sharp drops after each outer step (not to zero) followed by rapid return to steady state, consistent with partial consensus

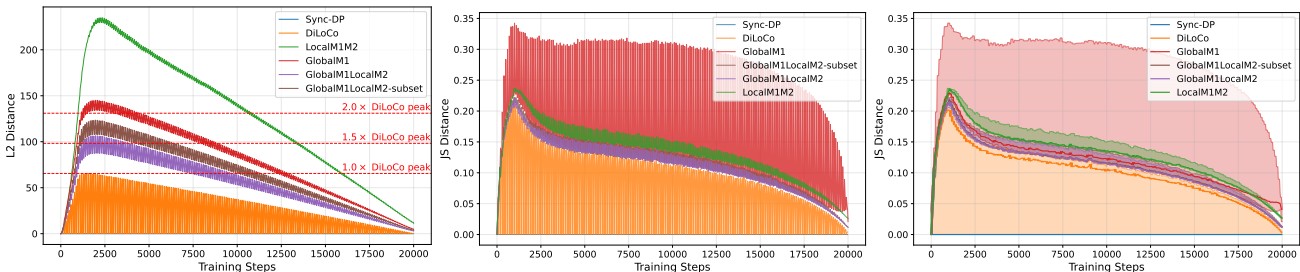

*Figure 4.* **Distance plots. (Left & Middle)** L2 and JS distance per step respectively. Note that Sync-DP has zero distance everywhere due to its global synchronization, and DiLoCo has zero at the end of every outer step (every $H = 100$ steps) due to the global synchronization there. **(Right)** Range (min and max) as well as median JS distance per step within a window of 125 steps. L2 distance aligns with our theory on consensus distance, while JS distance highlights instability that is reduced most by adding Mix2.

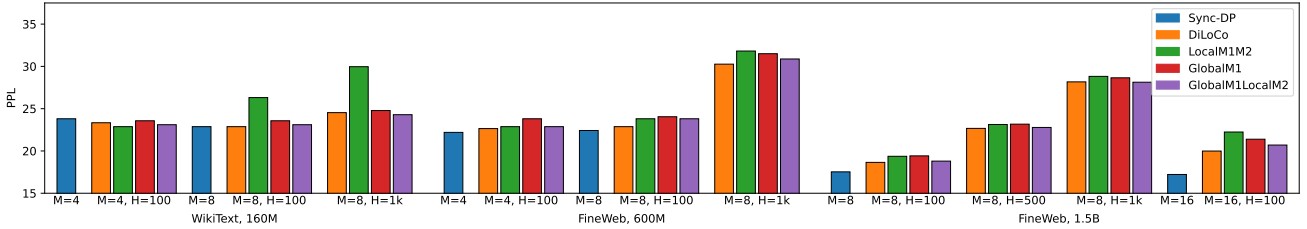

*Figure 5.* **Scaling ablations.** Validation perplexity for different number of workers ($M$) and inner steps ($H$) on three different model sizes, 160M parameters on WikiText, and 600M parameters on FineWeb, and 1.5B parameters on FineWeb. The ablations show a growing trade-off with scale: higher $M$ and $H$ amplify the cost of imperfect consensus, while stronger mixing mitigates the perplexity hit.

followed by local-step drift. JS distance behaves differently: it spikes *after* the outer step and then decays during the subsequent local steps. This suggests the outer update is the main source of short-term instability, while the local trajectory re-stabilizes the model.

L2 curves are cleanly separated and roughly matches the theory in with Section 4.1 peak near the end of warmup: DiLoCo is lowest; GlobalM1 reaches roughly $2\times$ DiLoCo; GlobalM1LocalM2 and GlobalM1LocalM2-subset sit between about $1.5\times$ and $2\times$; and LocalM1M2 is highest ($>2\times$). This matches the ordering predicted by Section 4.1.

JS curves are less separable but reveal stability effects within an outer step. Methods with local Mix2 (GlobalM1LocalM2 and LocalM1M2) show much smaller within-step variation, indicating Mix2 stabilizes the outer update. GlobalM1LocalM2 is slightly lower overall, reflecting the benefit of global non-blocking Mix1. GlobalM1 exhibits large drift within an outer step, yet its minimum within each outer step is comparable to GlobalM1LocalM2 and below LocalM1M2, suggesting global Mix1 improves the eventual (steady-state) functional consensus even when short-term instability is higher. The median plot highlights this steady state: all three methods in our formulation remain close to DiLoCo, with LocalM1M2 consistently highest and GlobalM1/GlobalM1LocalM2 similar for most of training (diverging late when GlobalM1's learning rate drops), consistent with their broadly similar validation performance.

## 6. Ablations

**Scaling.** In Figure 5 we evaluate two smaller Llama-style models (160M on WikiText (Merity et al., 2017) and 600M on FineWeb (Penedo et al., 2024)) while varying $M$ and $H$. On WikiText, Sync-DP is not always best, with DiLoCo and other low-communication variants sometimes slightly better, whereas on FineWeb Sync-DP remains strongest, consistent with our 1.5B results. For $H = 100$, performance is broadly similar across methods for both $M = 4$ and $M = 8$, except that LocalM1M2 degrades at $M = 8$ on WikiText, consistent with larger consensus error at higher worker counts. Increasing to $H = 1k$ has only a small effect at 160M, but is noticeably worse at 600M (and at 1.5B), indicating reduced tolerance to long local intervals as scale increases.

**Changing the number of inner steps** ($H = 100, 500, 1k$). Increasing $H$ raises compute utilization (Figure 3) but can stress stability for non-blocking GlobalM1. In Table 1, GlobalM1's max JS distance grows from 0.34 ($H = 100$) to 0.42 ($H = 500$) and 0.49 ($H = 1k$), with a corresponding (small) perplexity gap to DiLoCo at each $H$ (19.42 vs. 18.65; 23.17 vs. 22.67; 28.65 vs. 28.17 at 10B tokens). At the same time, GlobalM1 achieves much higher utilization, reaching 100% once Mix1 is fully hidden (already at $H = 500$ in our bandwidth range). Adding blocking communication via GlobalM1LocalM2-subset substantially reduces disagreement at $H = 100$ and $H = 500$ (max JS 0.22 and 0.28) while only slightly lowering utilization

(45%/82% at $H = 100$ and 92%/96% at $H = 500$), yielding a favorable utilization–stability trade-off compared to DiLoCo (36%/53%, 74%/85%).

**Increasing the number of workers to $M = 16$.** Table 1 includes two $M = 16$ blocks: *Global-BS-fix* (GBS=512, per-worker batch halves) and *Local-BS-fix* (GBS=1024, per-worker batch matched to the $M = 8$ baseline). All methods show a slight perplexity drop relative to $M = 8$, with the gap between non-blocking and blocking methods slightly widening, consistent with Charles et al. (2025) showing DiLoCo degrades with more workers and the longer convergence of mixing at larger $M$. Compute utilization is markedly higher under Local-BS-fix because the larger per-worker compute hides the all-reduce more effectively, while perplexity is somewhat worse since global batch is also larger.

## 7. Conclusion and Future Work

We introduce *Factored DiLoCo*, a restructuring of DiLoCo's outer synchronization that reduces blocking communication while preserving the key role of synchronization in reducing worker disagreement. Our approach factorizes synchronization into two mixing operations: a non-blocking Mix1 that can be overlapped with local computation without temporal staleness, and a blocking Mix2 that enforces stronger consensus when needed for stability. This factorization exposes a simple trade-off between compute utilization and optimization stability, where Mix1 provides a bounded level of consensus with minimal impact on utilization, and Mix2 tightens consensus at an explicit blocking cost.

In low-bandwidth regimes representative of consumer-grade decentralized settings, we empirically show that Factored DiLoCo substantially improves compute utilization with training progress ranging from comparable to closely matching DiLoCo across model scales and hyperparameter regimes. Most importantly, these gains reduce perplexity per unit wall-clock time, a metric on which decentralized training struggles (Figure 2, middle and right). We also observe more graceful degradation under transient communication failures, due to mixing-based synchronization avoiding having to abort and restart communication under failures. Beyond standard parameter-space disagreement, we find that functional discrepancy, measured by JS distance on logits, better tracks optimization instability, motivating its use for determining when to use blocking communication. We further use this metric to identify the parameter blocks driving consensus error, and verify in an ablation (Section D.3) that applying Mix2 only to these blocks recovers most of the full-Mix2 stability.

There are several promising directions for further developing the framework itself. First, we could determine whether to use Mix2 and with how many parameters automatically based on real-time signals (such as JS-distance spikes), rather than manually choosing such hyperparameters. Second, our analysis uses simplifying assumptions, thus extending the theory to more realistic assumptions could give better insights on how to navigate the compute utilization - optimization stability trade-off. Third, while we demonstrate improved robustness to transient failures, fully fault-tolerant decentralized training remains a challenge, especially under correlated delays and partial communication.

Beyond extending the framework itself, several orthogonal axes can also be incorporated to further reduce communication cost and/or improve consensus, and may compound gains when combined with our synchronization factorization. Pipeline parallelism amortizes bandwidth per stage and lets longer forward–backward step times hide more communication (Ryabinin et al., 2023; Ramasinghe et al., 2025), pipeline-parallel randomization reduces worker disagreement at the model-parallel level (Kolehmainen et al., 2025), parameter partitioning with per-partition communication schedules (Douillard et al., 2025) trades off bandwidth and consensus at finer granularity, and compression (quantization, sparsification, low-rank) reduces the payload of each communication round. The JS-distance metric could in particular guide where or how to apply compression.

## Impact Statement

This paper proposes a communication-efficient variant of DiLoCo that reduces blocking synchronization, improving the practicality of distributed and decentralized training in low-bandwidth and failure-prone settings.

As with most work that improves training efficiency, this may lower cost and infrastructure barriers and broaden access to large-model experimentation, but could also accelerate scaling and amplify downstream risks (e.g., misuse and unequal access). Our work introduces no new model capabilities, datasets, or user-facing systems, and does not address privacy, security, or incentives. Real deployments should apply standard access controls and safeguards.

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

# A. Factored Gossip DiLoCo: Detailed Algorithm

Algorithm 1 gives the full per-worker procedure for Factored Gossip DiLoCo. Each worker runs the algorithm in parallel. The two mixing operators Mix1 and Mix2 act on the corresponding parameter / outer-gradient values across workers; choosing them determines the framework variant (Table 2).

---

**Algorithm 1** Factored Gossip DiLoCo (per worker $m$)

---

**Require:** initial parameters $x_0$, outer rounds $R$, inner steps per round $H$, inner optimizer InnerOpt, outer optimizer OuterOpt, mixing operators Mix1 (non-blocking) and Mix2 (blocking)

1:   $z_{m,0} \leftarrow x_0$ {all workers start identical}
2:   **for** $r = 0, 1, \ldots, R-1$ **do**
3:     **launch** $x_{m,r} \leftarrow \text{Mix1}_r(z_{m,r})$ asynchronously {overlaps with the inner loop below}
4:     $y_{m,r,0} \leftarrow z_{m,r}$
5:     **for** $h = 0, 1, \ldots, H-1$ **do**
6:       sample minibatch $b_{m,r,h}$
7:       $g_{m,r,h} \leftarrow \nabla_y \mathcal{L}(y_{m,r,h}, b_{m,r,h})$
8:       $y_{m,r,h+1} \leftarrow \text{InnerOpt}(y_{m,r,h}, g_{m,r,h})$
9:     **end for**
10:    $\Delta_{m,r} \leftarrow y_{m,r,H} - y_{m,r,0}$ {outer pseudo-gradient}
11:    $\widehat{\Delta}_{m,r} \leftarrow \text{Mix2}_r(\Delta_{m,r})$ {blocking}
12:    **wait** for $\text{Mix1}_r$ to complete; $x_{m,r}$ is its result
13:    $z_{m,r+1} \leftarrow \text{OuterOpt}\left(x_{m,r}, -\widehat{\Delta}_{m,r}\right)$
14: **end for**
15: **return** $\overline{z}_R = \frac{1}{M} \sum_{m=1}^{M} z_{m,R}$ {final global model}

---

**Inner / outer optimizers.** In all our 1.5B experiments, InnerOpt is AdamW with peak inner learning rate $3 \times 10^{-4}$, 1000-step linear warmup, and linear decay to zero. OuterOpt is SGD with Nesterov momentum (outer learning rate 0.7, momentum 0.9), matching DiLoCo's recipe. Smaller-model scaling ablations (Figure 5) use inner learning rate $4 \times 10^{-4}$.

**Choosing Mix1 and Mix2.** The four main framework configurations in the body correspond to four (Mix1, Mix2) pairs (Table 2); the GlobalM1S variants additionally apply Mix1-style averaging to the outer optimizer state (see Table 3). For the GlobalM1LocalM2-subset variant, Mix2 is applied only to a chosen subset of parameter blocks (the rest pass through identity); the subset selection methodology is in Section D.3.

*Table 2.* The four main framework configurations realized as (Mix1, Mix2) choices in Algorithm 1. "All-reduce" is a global average across workers; "pairwise gossip" is one round of random-matching pairwise averaging; "identity" leaves the input unchanged.

| Configuration | Mix1 | Mix2 |
|---|---|---|
| DiLoCo | identity | global all-reduce |
| LocalM1M2 | pairwise gossip | pairwise gossip |
| GlobalM1 | global all-reduce | identity |
| GlobalM1LocalM2 | global all-reduce | pairwise gossip |

# B. Further Results

In Section 4.2 we assumed that the outer optimizer is linear in the gradient and has global parameters. So the question becomes whether the outer optimizer that DiLoCo uses, SGD with Nesterov momentum, is linear in the gradient and has global parameters as per our assumption. The first part holds for the implementation version of SGD with Nesterov

*Table 3.* Further ablations with $M = 8$ workers. Here, *local* denotes pairwise gossip averaging and *global* denotes full averaging across all replicas.

| Config (row) | H | Mix1 | Mix1 state | Mix2 | Val PPL at $T$ tokens 10B | 30B | Compute Util. at 100Mbps | 200Mbps | Max Distance L2 | JS |
|---|---|---|---|---|---|---|---|---|---|---|
| **GlobalM1S** | 100 | global | global | none | 20.91 | - | 28% | 56% | 125.5 | 1.13 |
| **GlobalM1SLocalM2** | 100 | global | global | local | 19.11 | - | 22% | 43% | 93.3 | 0.25 |
| **GlobalM1SLocalM2-subset** | 100 | global | global | local | 19.69 | - | 25% | 50% | 106.1 | 0.26 |

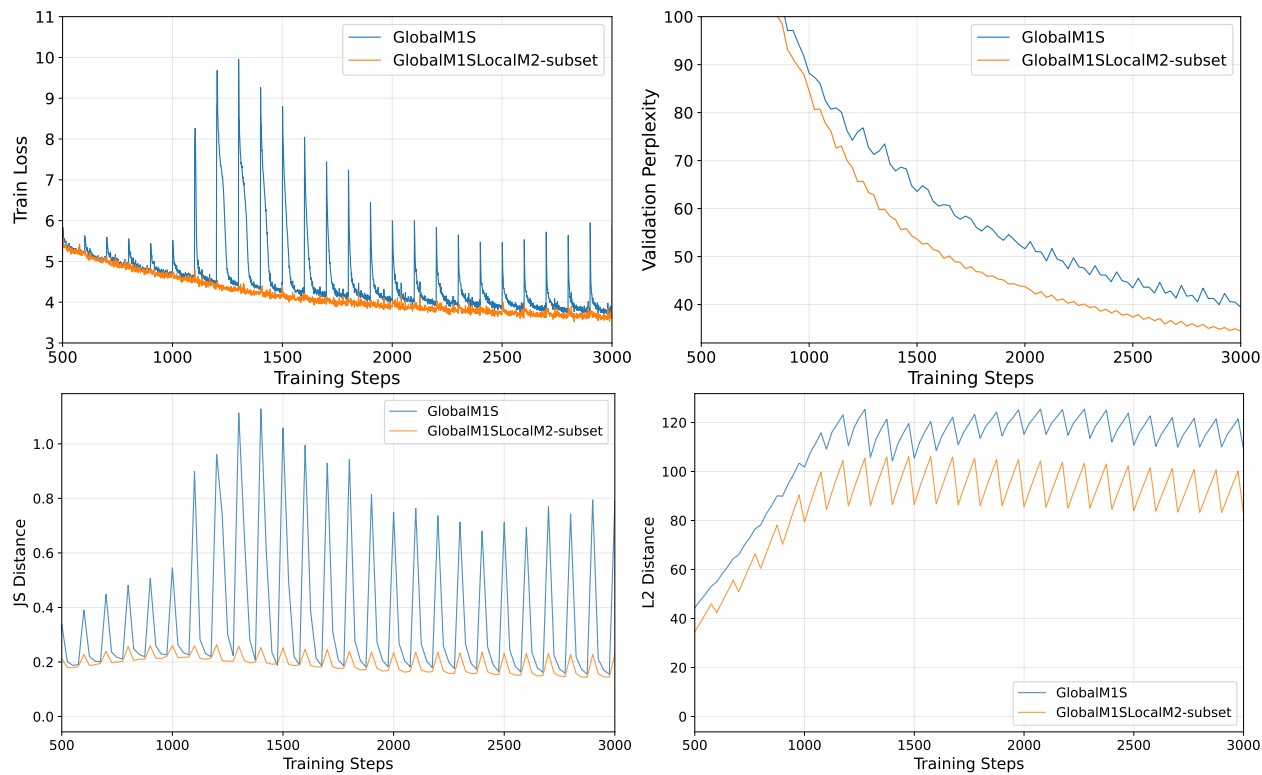

*Figure 6.* Example of training instability with Global Mix1 and no Mix 2, and then adding Mix2 with only a subset of the parameters to stabilize it. Also note that JS distance shows instability, unlike L2 distance.

momentum, whose update step is implemented in PyTorch (Paszke et al., 2019) as

$$v_{m,r+1} = \mu v_{m,r} + g_{m,r} \tag{36}$$
$$\text{(Momentum Update)}$$

$$x_{m,r+1} = x_r - \gamma \left( g_{m,r} + \mu v_{m,r+1} \right) \tag{37}$$
$$\text{(Optimization Step)}$$

$$\therefore x_{m,r+1} = x_r - \gamma \left( (1+\mu) g_{m,r} + \mu^2 v_{m,r} \right). \tag{38}$$

However, for the optimizer parameters to be global, we need to add an extra global averaging of the momentum $v_{m,r}$ first. Note that this is still not blocking and can be considered an extra part of Mix1. This motivates us to consider the following methods

- **GlobalM1S**: our framework with global averaging for Mix1 that also includes outer optimizer state, and no Mix2, thus no blocking communication.
- **GlobalM1SLocalM2**: our framework with global averaging for Mix1 that also includes outer optimizer state, and pairwise gossip for Mix2.

We now give results with this.

**Synchronizing outer optimizer state.** Unexpectedly, as shown by the results in the second section of Table 1, additionally synchronising the outer optimizer state make our formulation perform slightly worse. In particular, it made the optimization of our non-blocking method (GlobalM1S) much more unstable, with large training loss and validation loss spikes, thus requiring the optimization to recover from it. We did not observe this at lower parameter scales. Incidentally, we found extremely large JS-distance spikes coincide with the loss spikes. Adding blocking communication, whether it be the full outer gradients with GlobalM1SLocalM2 or a subset of them with GlobalM1SLocalM2-subset, drastically reduced the magnitude of any such loss spikes, as well as the JS-distance spikes. This can be seen in the max JS distance, which falls from 1.13 to 0.25/0.26, which is in the range of the main experiments. We also show zoomed in plots of this in Figure 6. Thus, these experiments show that keeping track of the JS-distance is a good way to monitor how stable the optimization is, and that to stabilise the optimization blocking communication should be added.

## B.1. Comparison with NoLoCo

We compare against a reimplementation of NoLoCo that includes its full outer step (including dampening) but excludes pipeline randomization. Pipeline-parallel randomization is orthogonal to the DP-only setting we focus on and would conflate this comparison with a different axis, so we exclude it to be consistent with how LocalM1M2 is framed in Section 4.1.

We evaluate two hyperparameter configurations:

- **DiLoCo-style HPs**, matching our other 1.5B runs: outer LR 0.7, outer momentum 0.9, $H = 100$, inner LR $3 \times 10^{-4}$, with dampening tied to the outer LR per NoLoCo's official code. This run *diverged after approximately 5k steps*.
- **NoLoCo paper HPs**, matching their reported 1.3B setup: outer momentum 0.5, $H = 50$, inner LR $2 \times 10^{-4}$, dampening per their code. This run converged, reaching a validation perplexity of 22.65 at 10B tokens.

The converged result is substantially worse than DiLoCo and all four of our framework variants (Table 4), *despite* using $H = 50$ rather than $H = 100$. Note that halving $H$ should reduce the consensus distance and thus improve convergence (at the cost of twice the communication), not degrade it. This suggests pipeline randomization is important for both their performance and stability, though their method still appears unstable.

*Table 4.* NoLoCo (10B-token validation perplexity) against DiLoCo and our framework variants. NoLoCo's paper-HP configuration uses $H = 50$, which is twice the communication of the $H = 100$ baselines.

| Method | H | Val PPL (10B) |
|---|---|---|
| NoLoCo (paper HPs) | 50 | 22.65 |
| NoLoCo (DiLoCo HPs) | 100 | diverged ($\sim$5k steps) |
| DiLoCo | 100 | 18.65 |
| LocalM1M2 | 100 | 19.37 |
| GlobalM1 | 100 | 19.42 |
| GlobalM1LocalM2 | 100 | 18.80 |

Upon reinspection of their formulation, we note that NoLoCo's outer step is described as "modified Nesterov momentum" but lacks the lookahead step of true Nesterov momentum. Prior work on outer optimizers for pseudo-gradient methods (Douillard et al., 2023; Kallusky et al., 2025) shows the lookahead is critical: Kallusky et al. (2025) in particular isolates this effect and reports a substantial gap between Nesterov-with-lookahead and the "modified Nesterov" variant. We verified that NoLoCo's official code matches the description in their paper, so we attribute the gap primarily to the missing lookahead rather than to an implementation difference, though we did not confirm this experimentally.

## B.2. Comparison with Streaming DiLoCo's Overlap

Streaming DiLoCo (Douillard et al., 2025) has three components: parameter partitioning, computation–communication overlap, and quantization. We reimplement only the computation–communication overlap component for our comparison, since that is the only part that is directly comparable. Both parameter partitioning (with its own communication schedule) and quantization are orthogonal to our framework and could be added on top. Due to minor ambiguities in the published pseudocode and the lack of released code, we corresponded with the Streaming DiLoCo authors to confirm implementation details.

We note that these results are in the appendix rather than the main paper for two reasons. First, the overlap mechanism introduces temporal staleness, placing it in a category of methods we do not compare against in the main paper (see Section 4.3). Second, the overlap mechanism yields negligible compute-utilization gains over DiLoCo at our target low-bandwidth settings. Streaming DiLoCo recommends a maximum overlap of 5 steps, since performance degrades with longer overlap (their Figs. 8 and 10). This 5-step ceiling is enough to substantially improve compute utilization (CU) over DiLoCo in their high-bandwidth setting, from 85% to 95% at 10Gbps, and to 100% when combined with quantization. In our low-bandwidth setting, however, the same 5-step overlap yields essentially no CU improvement over DiLoCo: $36\% \rightarrow 36\%$ at 100Mbps and $53\% \rightarrow 54\%$ at 200Mbps. By contrast, our methods substantially improve CU at these bandwidths (Figure 3).

We evaluated two hyperparameter configurations:

- **5-step overlap, DiLoCo-style HPs** matching our other 1.5B runs ($H = 100$, DiLoCo outer hyperparameters): large validation-loss spikes during merging, *diverged after ~900 steps*.
- **5-step overlap, Streaming DiLoCo paper HPs** ($H = 30$, outer LR 0.4): smaller spikes, but *still diverged after ~500 steps* (which is many more outer rounds in absolute terms, since $H = 30$ means more outer rounds per training step).

These experiments are a clear example of how JS distance is more informative than L2 distance, complementing our observations in Figure 4 and Section D.3. At the point of divergence the L2 consensus distance was small (peak $\approx 22.4$), comparable to the L2 distances of our stable runs (Figure 4). The JS consensus distance, however, spiked sharply, reaching peaks of approximately 3. This is an order of magnitude above the $\sim 0.3$–$0.4$ peaks seen on our stable methods.

These results suggest that Streaming DiLoCo's parameter partitioning is crucial for providing robustness to staleness and stabilizing the optimization. Our method appears more stable, as it converges with the original DiLoCo paper's hyperparameters and provides meaningful compute utilization gains in low-bandwidth settings.

## C. Further Discussion

**Potential Benefits of Worker Disagreement.** Kong et al. (2021) also show that (1) the consensus distance is especially sensitive for initial phase of training, (2) reducing it significantly lower than the critical consensus distance does not result in better performance gains, and (3) maintaining a large consensus distance in later training phases can be beneficial. Other papers (Zhu et al., 2023) have also shown that having non-zero consensus distance can in fact improve performance over the globally averaging version due to it acting like sharpness aware minimization (Foret et al., 2021), where the randomness of the worker parameters around the true average of the workers' parameters ensure that the overall average model converges to flatter minima.

**Other ways to control consensus.** As discussed above, we can add blocking communication to improve consensus via Mix2, and we can trade-off between consensus and compute utilization by adjusting both the amount of mixing and the percentage of parameters in Mix2. Another way to flexibly add blocking communication is to introduce sparse weight averaging during the local steps, such as SPARTA (Beton et al., 2025). In SPARTA, at each step a sparse random subset of the parameters (e.g., 0.05% of the parameters) are globally averaged. They show that integrating their method into DiLoCo helps reduce consensus between workers, enabling a larger number of inner steps to be stable. This could easily be integrated into our framework as well.

**Preferred method per bandwidth.** Combining the perplexity gap (Table 1) with the utilization curves (Figure 3) gives a simple rule of thumb at $H = 100$, $M = 8$, summarized in Table 5: DiLoCo when bandwidth is plentiful, GlobalM1LocalM2 when blocking communication is still affordable, the subset variant in the intermediate range (or other compression variants which we did not investigate), and GlobalM1 when bandwidth is the binding constraint.

## D. Consensus and Functional Disagreement Metrics

In the main paper we track two complementary notions of worker disagreement: an L2-based *parameter disagreement* and a Jensen–Shannon (JS) based *functional disagreement*. Both are computed at a fixed training time (e.g., at a given inner step $h$ of outer round $r$) over the set of $M$ workers.

*Table 5.* Recommended configuration per bandwidth regime at $H = 100$, $M = 8$, balancing perplexity (Table 1) against compute utilization (Figure 3). GlobalM1LocalM2-subset is a diagnostic-driven configuration (that represents the possibility of compression integration) rather than a peer of the four main framework variants, see Section 4.4.

| Recommended | Bandwidth |
|---|---|
| DiLoCo | $\geq 2\,\text{Gbps}$ |
| GlobalM1LocalM2 | $500\,\text{Mbps} - 2\,\text{Gbps}$ |
| GlobalM1LocalM2-subset | $100 - 500\,\text{Mbps}$ |
| GlobalM1 | $< 100\,\text{Mbps}$ |

### D.1. L2 Parameter Disagreement

Let $x_{m,r,h} \in \mathbb{R}^d$ denote the parameter vector on worker $m \in \{1, \ldots, M\}$ at outer round $r$ and inner step $h$. Define the global average parameter

$$\overline{x}_{r,h} := \frac{1}{M} \sum_{m=1}^{M} x_{m,r,h}. \tag{39}$$

We report an L2 consensus distance using the mean-squared deviation from the global average,

$$CD_{L2}(X_{r,h}) := \frac{1}{M} \sum_{m=1}^{M} \|x_{m,r,h} - \overline{x}_{r,h}\|_2^2. \tag{40}$$

### D.2. JS Functional Disagreement

Parameter disagreement does not always reflect whether workers make similar predictions. We therefore measure *functional* disagreement using the (normalized) Jensen–Shannon (JS) distance between each worker's predicted token distribution and that of the *parameter-averaged* model.

Let $x_{m,r,h} \in \mathbb{R}^d$ denote worker $m$'s parameters at outer round $r$ and inner step $h$, and define the parameter average

$$\overline{x}_{r,h} := \frac{1}{M} \sum_{m=1}^{M} x_{m,r,h}. \tag{41}$$

For an input example $u$ (e.g., a token position in a held-out batch), let $z(x, u) \in \mathbb{R}^K$ be the model logits and

$$p(x, u) := \text{softmax}(z(x, u)) \in \Delta^{K-1} \tag{42}$$

the corresponding categorical distribution. We define the worker and averaged-model distributions as

$$p_{m,r,h}(u) := p(x_{m,r,h}, u), \qquad \overline{p}_{r,h}(u) := p(\overline{x}_{r,h}, u). \tag{43}$$

For each example $u$, we compute the JS divergence between two distributions

$$\text{JS}\big(p_{m,r,h}(u), \overline{p}_{r,h}(u)\big) := \frac{1}{2}\text{KL}\left(p_{m,r,h}(u) \,\middle\|\, \frac{p_{m,r,h}(u) + \overline{p}_{r,h}(u)}{2}\right) + \frac{1}{2}\text{KL}\left(\overline{p}_{r,h}(u) \,\middle\|\, \frac{p_{m,r,h}(u) + \overline{p}_{r,h}(u)}{2}\right), \tag{44}$$

and report the *normalized JS distance*

$$d_{\text{JS}}\big(p_{m,r,h}(u), \overline{p}_{r,h}(u)\big) := \sqrt{\frac{\text{JS}\big(p_{m,r,h}(u), \overline{p}_{r,h}(u)\big)}{\log 2}}. \tag{45}$$

Finally, we average over workers and a fixed batch $\mathcal{U}$ to obtain a scalar metric:

$$CD_{JS}(X_{r,h}) := \frac{1}{M|\mathcal{U}|} \sum_{m=1}^{M} \sum_{u \in \mathcal{U}} d_{\text{JS}}\big(p_{m,r,h}(u), \overline{p}_{r,h}(u)\big). \tag{46}$$

Implementation wise, it is important to use float64 and do everything in the log domain. For efficiency we only compute the distance on the 100 tokens with the highest probability in the parameter averaged model, where the top 100 recomputed every time, as otherwise JS distance is computationally and GPU memory intensive.

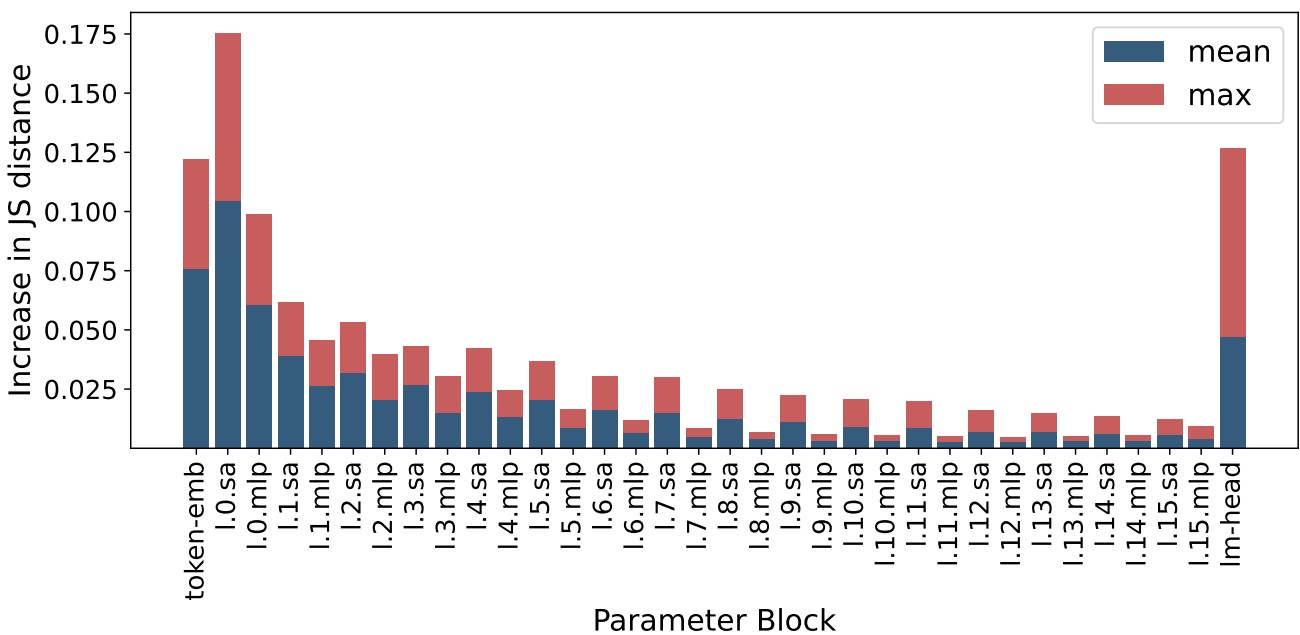

*Figure 7.* **Parameter block consensus sensitivity.** JS distance increase for each parameter block when only that block is averaged, taking the mean and the max over the course of training. Token embeddings and the LM head dominate, followed by self-attention and then MLP parameters, with the latter two decreasing with depth.

### D.3. JS-Distance Metric and Subset Selection

In Section 4.4 we introduced the GlobalM1LocalM2-subset variant as a controlled experiment to test whether the JS-distance metric identifies useful parameter blocks for Mix2. Here we describe how that subset is selected and report a subset-length ablation.

**Per-block sensitivity analysis.** Because JS distance is a functional metric, it can be probed per parameter block. For each block in the model we measure the average JS-distance increase per worker when *only* that block is averaged during optimization, performed once offline per model architecture (no impact on training compute). Figure 7 reports the mean and the max of this sensitivity over the course of training.

Token embeddings and the LM head dominate, followed by self-attention parameters and then MLP parameters, with the latter two decreasing with depth. This naturally suggests a subset comprised of token embeddings, the LM-Head, and the first $k$ transformer layers.

**Choosing $k$.** Since the goal is to suppress peak JS-distance spikes, we sweep $k$ and measure the peak JS distance through training, with results in Table 6. We use $k = 2$ (43% of parameters for our 1.5B model), as that is sufficient to bring peak JS distance close to GlobalM1LocalM2 levels.

*Table 6.* Subset-length ablation. Peak JS distance through training as the subset is grown to include progressively more transformer layers.

| Subset | Peak JS distance |
|---|---|
| Token Emb + LM-Head | 0.28 |
| + Layer 0 | 0.26 |
| + Layer 1 | 0.22 |

The subset-length sweep, the per-block sensitivity (Figure 7), and the configuration's main-paper results (Table 1 and Figures 2 to 4) together suggest that the JS-distance metric is useful in determining where Mix2 communication is actually needed.

# E. Consensus Error Result and Proof

We structure our proof based on Khaled et al. (2025).

Let us rewrite our formulation with SGD

$$x_{m,r} = \text{Mix1}_r\left(z_{m,r}\right) \tag{47}$$

$$y_{m,r,0} = z_{m,r} \tag{48}$$

for $h = 0, 1, \ldots, H-1$ in sequence:

$$g_{m,r,h} = \nabla_{y_{m,r,h}} \mathcal{L}(y_{m,r,h}, b_{m,r,h}) \tag{49}$$

$$y_{m,r,h+1} = y_{m,r,h} - \eta g_{m,r,h} \tag{50}$$

$$\Delta_{m,r} = y_{m,r,H} - y_{m,r,0} = -\eta \sum_{h=0}^{H-1} g_{m,r,h} \tag{51}$$

$$\widehat{\Delta_{m,r}} = \text{Mix2}_r\left(\Delta_{m,r}\right) \tag{52}$$

$$z_{m,r+1} = x_{m,r} + \widehat{\Delta}_{m,r} \tag{53}$$

Following Khaled et al. (2025), we define

$$y_{r,h} \overset{\text{def}}{=} \frac{1}{M}\sum_{m=1}^{M} y_{m,r,h}, \qquad g_{r,h} \overset{\text{def}}{=} \frac{1}{M}\sum_{m=1}^{M} g_{m,r,h}$$

$$\overline{g}_{m,r,h} \overset{\text{def}}{=} \mathbb{E}_{r,h-1}\left[g_{m,r,h}\right] = \nabla f(y_{m,r,h}), \qquad \overline{g}_{r,h} \overset{\text{def}}{=} \mathbb{E}_{r,h-1}\left[g_{r,h}\right] \tag{54}$$

and noise $n_{m,r,h} = g_{m,r,h} - \overline{g}_{m,r,h}$.

We also define $\mathcal{F}_{r,h}$ as the $\sigma$-algebra generated by all the stochastic gradients up to the start of (but not including) step $h$ in round $r$, and will often take the conditional expectation on $\mathcal{F}_{r,h}$, i.e., $\mathbb{E}\left[\cdot \mid \mathcal{F}_{r,h}\right]$.

Assume $\rho_1, \rho_2$ are such that

$$\|\text{Mix1}(X) - \overline{X}\|_F^2 \le \rho_1 \|X - \overline{X}\|_F^2 \tag{55}$$

$$\|\text{Mix2}(X) - \overline{X}\|_F^2 \le \rho_2 \|X - \overline{X}\|_F^2. \tag{56}$$

We have the following assumptions

**Assumption 1** The function $f$ is differentiable, convex, has $L$-Lipschitz gradients, and has a minimizer $x_*$.

**Assumption 2** Given a point $x \in \mathbb{R}^d$, the stochastic gradients $g(x) \in \mathbb{R}^d$ are (a) unbiased in expectation $\mathbb{E}\left[g(x)\right] = \nabla f(x)$, and (b) has variance bounded as $\mathbb{E}\left[\|g(x) - \nabla f(x)\|^2\right] \le \sigma^2$, where $\mathbb{E}\left[\cdot\right]$ denotes the expectation operator.

**Assumption 3** Minibatches (and hence stochastic gradients) are sampled i.i.d. across workers and steps.

**Assumption 4** There exists $\mu_r$ (measurable w.r.t. $\mathcal{F}_{r,0}$) such that $\mathbb{E}\left[\Delta_{m,r} \mid \mathcal{F}_{r,0}\right] = \mu_r$ for all $m$. Equivalently, $\mathbb{E}\left[\Delta_{m,r} - \Delta_{s,r} \mid \mathcal{F}_{r,0}\right] = 0$ for any $m, s \in \{1, ..., M\}$.

The first two (and implicitly the third) are used in Khaled et al. (2025). We further add Assumption 4, which essentially says that even though we have different starting points at the start of our outer step, they are following a common mean direction. This comes for free in DiLoCo due to the global synchronization. This assumption can also be thought of as a stabilizing assumption on the optimization.

We also restate the following Lemma from Khaled et al. (2025) (proof is given in that paper):

**Lemma E.1.** *Let $f$ be a convex and $L$-smooth function. Suppose that $\eta \leq \frac{2}{L}$, and let*

$$T_\eta(x) = x - \eta \nabla f(x).$$

*Then,*

$$\|T_\eta(x) - T_\eta(y)\|^2 \leq \|x - y\|^2.$$

We now state our lemma.

**Lemma E.2.** *Suppose Assumptions 1–4 hold and $\eta \leq \frac{1}{L}$. Then for $\rho_1 \in [0, 1)$ and all $r, h$,*

$$\mathbb{E}\left[\frac{1}{M^2} \sum_{m,s=1}^{M} \|y_{m,r,h} - y_{s,r,h}\|^2\right] \leq 2\eta^2 \sigma^2 H \left(1 + \frac{\rho_2}{1 - \rho_1}\right).$$

*The boundary case $\rho_1 = 1$ (vanilla DiLoCo) is treated in Theorem E.3 below.*

*Proof.* We first show that

$$\mathbb{E}\left[\mathcal{V}_{r,h}\right] \leq \mathbb{E}\left[\mathcal{V}_{r,0}\right] + 2\eta^2 \sigma^2 h. \tag{57}$$

Let $\tilde{T}_\eta(y_{m,r,h}) = y_{m,r,h} - \eta g_{m,r,h}$ where $g_{m,r,h}$ is the stochastic gradient, and $T_\eta(y_{m,r,h}) = y_{m,r,h} - \eta \bar{g}_{m,r,h}$ is the corresponding expected gradient update. Their difference is the noise term $\xi_{m,r,h} = \tilde{T}_\eta(y_{m,r,h}) - T_\eta(y_{m,r,h}) = -\eta n_{m,r,h}$. Thus

$$y_{m,r,h+1} - y_{s,r,h+1} = \tilde{T}_\eta(y_{m,r,h}) - \tilde{T}_\eta(y_{s,r,h}) = T_\eta(y_{m,r,h}) - T_\eta(y_{s,r,h}) + [\xi_{m,r,h} - \xi_{s,r,h}].$$

Define $\mathcal{V}_{r,h} = \frac{1}{M^2} \sum_{m,s=1}^{M} \|y_{m,r,h} - y_{s,r,h}\|^2$. It follows that

$$\begin{aligned}
\mathcal{V}_{r,h+1} &= \frac{1}{M^2} \sum_{m,s=1}^{M} \|y_{m,r,h+1} - y_{s,r,h+1}\|^2 \\
&= \frac{1}{M^2} \sum_{m,s=1}^{M} \left[ \|T_\eta(y_{m,r,h}) - T_\eta(y_{s,r,h})\|^2 + \|\xi_{m,r,h} - \xi_{s,r,h}\|^2 \right. \\
&\qquad\qquad \left. + 2 \langle T_\eta(y_{m,r,h}) - T_\eta(y_{s,r,h}), \xi_{m,r,h} - \xi_{s,r,h} \rangle \right].
\end{aligned}$$

Taking the conditional expectation on $\mathcal{F}_{r,h}$ gives

$$\mathbb{E}\left[\mathcal{V}_{r,h+1} \mid \mathcal{F}_{r,h}\right] = \frac{1}{M^2} \sum_{m,s=1}^{M} \left[ \|T_\eta(y_{m,r,h}) - T_\eta(y_{s,r,h})\|^2 + \mathbb{E}\left[\|\xi_{m,r,h} - \xi_{s,r,h}\|^2 \mid \mathcal{F}_{r,h}\right] \right]$$

where we drop $\mathbb{E}\left[\langle T_\eta(y_{m,r,h}) - T_\eta(y_{s,r,h}), \xi_{m,r,h} - \xi_{s,r,h} \rangle \mid \mathcal{F}_{r,h}\right]$ because the left part is fixed given the conditioning $\sigma$-algebra and since the stochastic gradients are unbiased $\mathbb{E}\left[\xi_{m,r,h} \mid \mathcal{F}_{r,h}\right] = \mathbb{E}\left[\xi_{s,r,h} \mid \mathcal{F}_{r,h}\right] = 0$.

Finally, using the fact that $\|T_\eta(x) - T_\eta(y)\|^2 \leq \|x - y\|^2$ whenever $\eta \leq \frac{2}{L}$ from Theorem E.1, as well as Assumption 2 and Assumption 3, we get

$$\begin{aligned}
\mathbb{E}_{r,h}\left[\mathcal{V}_{r,h+1}\right] &\leq \frac{1}{M^2} \sum_{m,s=1}^{M} \left[ \|y_{m,r,h} - y_{s,r,h}\|^2 + 2\eta^2 \sigma^2 \right] \\
&= \mathcal{V}_{r,h} + 2\eta^2 \sigma^2
\end{aligned}$$

Thus, taking the unconditional expectation and recursing from $h = 0$ we get

$$\mathbb{E}\left[\mathcal{V}_{r,h}\right] \leq \mathbb{E}\left[\mathcal{V}_{r,0}\right] + 2\eta^2 \sigma^2 h. \tag{58}$$

We now need to determine $\mathbb{E}\left[\mathcal{V}_{r,0}\right]$. Note that

$$\|z_{m,r+1} - z_{s,r+1}\|^2 = \left\|(x_{m,r} - x_{s,r}) + \left(\widehat{\Delta_{m,r}} - \widehat{\Delta_{s,r}}\right)\right\|^2 \tag{59}$$

$$= \|x_{m,r} - x_{s,r}\|^2 + \left\|\widehat{\Delta_{m,r}} - \widehat{\Delta_{s,r}}\right\|^2 + 2\left\langle x_{m,r} - x_{s,r}, \widehat{\Delta_{m,r}} - \widehat{\Delta_{s,r}}\right\rangle. \tag{60}$$

Conditioning on $\mathcal{F}_{r,0}$ and using Assumption 4 and linearity of Mix2

$$\mathbb{E}\left[\widehat{\Delta_{m,r}} - \widehat{\Delta_{s,r}} \mid \mathcal{F}_{r,0}\right] = \mathbb{E}\left[\text{Mix2}\left(\Delta_{m,r} - \Delta_{s,r}\right) \mid \mathcal{F}_{r,0}\right] \tag{61}$$

$$= \text{Mix2}\left(\mathbb{E}\left[\Delta_{m,r} - \Delta_{s,r} \mid \mathcal{F}_{r,0}\right]\right) \tag{62}$$

$$= 0. \tag{63}$$

Thus as $x_{m,r} - x_{s,r}$ is $\mathcal{F}_{r,0}$ measurable,

$$\mathbb{E}\left[\left\langle x_{m,r} - x_{s,r}, \widehat{\Delta_{m,r}} - \widehat{\Delta_{s,r}}\right\rangle \mid \mathcal{F}_{r,0}\right] = \left\langle x_{m,r} - x_{s,r}, \mathbb{E}\left[\widehat{\Delta_{m,r}} - \widehat{\Delta_{s,r}} \mid \mathcal{F}_{r,0}\right]\right\rangle = 0. \tag{64}$$

Therefore

$$\mathbb{E}\left[\|z_{m,r+1} - z_{s,r+1}\|^2 \mid \mathcal{F}_{r,0}\right] = \|x_{m,r} - x_{s,r}\|^2 + \mathbb{E}\left[\left\|\widehat{\Delta_{m,r}} - \widehat{\Delta_{s,r}}\right\|^2 \mid \mathcal{F}_{r,0}\right]. \tag{65}$$

Now

$$\mathbb{E}\left[\mathcal{V}_{r+1,0}\right] = \mathbb{E}\left[\frac{1}{M^2}\sum_{m,s=1}^{M}\left[\|x_{m,r} - x_{s,r}\|^2\right]\right] + \mathbb{E}\left[\frac{1}{M^2}\sum_{m,s=1}^{M}\left[\left\|\widehat{\Delta_{m,r}} - \widehat{\Delta_{s,r}}\right\|^2\right]\right] \tag{66}$$

$$\leq \rho_1 \mathbb{E}\left[\frac{1}{M^2}\sum_{m,s=1}^{M}\left[\|z_{m,r} - z_{s,r}\|^2\right]\right] + \rho_2 \mathbb{E}\left[\frac{1}{M^2}\sum_{m,s=1}^{M}\left[\|\Delta_{m,r} - \Delta_{s,r}\|^2\right]\right] \tag{67}$$

$$\leq \rho_1 \mathbb{E}\left[\mathcal{V}_{r,0}\right] + \rho_2 D_r \tag{68}$$

$$\tag{69}$$

where $D_r := \mathbb{E}\left[\frac{1}{M^2}\sum_{m,s=1}^{M}\left[\|\Delta_{m,r} - \Delta_{s,r}\|^2\right]\right]$.

We now show that $D_r \leq 2\eta^2\sigma^2 H$.

We define the gradient noise, and note its properties from Assumption 2

$$g_{m,r,h} = \nabla f(y_{m,r,h}) + \xi_{m,r,h}, \quad \mathbb{E}[\xi_{m,r,h} \mid y_{m,r,h}] = 0, \quad \mathbb{E}[\|\xi_{m,r,h}\|^2 \mid y_{m,r,h}] \leq \sigma^2. \tag{70}$$

Then

$$\Delta_{m,r} = -\eta\sum_{h=0}^{H-1}\nabla f(y_{m,r,h}) - \eta\sum_{h=0}^{H-1}\xi_{m,r,h}. \tag{71}$$

Define

$$\varepsilon_{m,r} := \Delta_{m,r} - \mathbb{E}[\Delta_{m,r} \mid \mathcal{F}_{r,0}], \tag{72}$$

so $\mathbb{E}[\varepsilon_{m,r} \mid \mathcal{F}_{r,0}] = 0$, $\Delta_{m,r} - \Delta_{s,r} = \varepsilon_{m,r} - \varepsilon_{s,r}$ and $\varepsilon_{m,r} = -\eta\sum_{h=0}^{H-1}\xi_{m,r,h}$.

Conditioning on $\mathcal{F}_{r,0}$

$$\mathbb{E}[|\varepsilon_{m,r}|^2 \mid \mathcal{F}_{r,0}] = \eta^2\sum_{h=0}^{H-1}\mathbb{E}[|\xi_{m,r,h}|^2 \mid \mathcal{F}_{r,0}] \leq \eta^2 H\sigma^2. \tag{73}$$

so by independence across workers for $m \neq s$

$$\mathbb{E}\big[\langle \varepsilon_{m,r}, \varepsilon_{s,r}\rangle \mid \mathcal{F}_{r,0}\big] = 0 \tag{74}$$

and

$$\mathbb{E}\big[\|\Delta_{m,r} - \Delta_{s,r}\|^2 \mid \mathcal{F}_{r,0}\big] = \mathbb{E}\big[\|\varepsilon_{m,r} - \varepsilon_{s,r}\|^2 \mid \mathcal{F}_{r,0}\big] = \mathbb{E}\big[\|\varepsilon_{m,r}\|^2\big] + \mathbb{E}\big[\|\varepsilon_{s,r}\|^2\big] \leq 2\eta^2 H\sigma^2. \tag{75}$$

Averaging over $m$ and $s$ gives

$$D_r = \mathbb{E}\left[\frac{1}{M^2}\sum_{m,s}|\Delta_{m,r} - \Delta_{s,r}|^2\right] \leq 2\eta^2\sigma^2 H. \tag{76}$$

We now have the recursion

$$\mathbb{E}[V_{r+1,0}] \leq \rho_1 \mathbb{E}[V_{r,0}] + 2\rho_2\eta^2\sigma^2 H, \tag{77}$$

and since $V_{0,0} = 0$, for $\rho_1 < 1$ the geometric series converges and

$$\mathbb{E}[V_{r,0}] \leq \frac{2\rho_2\eta^2\sigma^2 H}{1 - \rho_1}. \tag{78}$$

Thus for $\rho_1 < 1$,

$$\mathbb{E}[V_{r,h}] \leq \frac{2\rho_2\eta^2\sigma^2 H}{1 - \rho_1} + 2\eta^2\sigma^2 h \tag{79}$$

$$\leq \frac{2\rho_2\eta^2\sigma^2 H}{1 - \rho_1} + 2\eta^2\sigma^2 H \tag{80}$$

$$= 2\eta^2\sigma^2 H\left(1 + \frac{\rho_2}{1 - \rho_1}\right). \tag{81}$$

$\square$

*Remark* E.3 (Boundary case $\rho_1 = 1$). The bound assumes $\rho_1 < 1$ so the geometric series converges. For vanilla DiLoCo ($\rho_1 = 1$, $\rho_2 = 0$), the recursion becomes $\mathbb{E}[V_{r+1,0}] \leq \mathbb{E}[V_{r,0}]$ with $\mathbb{E}[V_{0,0}] = 0$, so $\mathbb{E}[V_{r,0}] = 0$ for all $r$, and $\mathbb{E}[V_{r,h}] \leq 2\eta^2\sigma^2 H$, recovering the result of Khaled et al. (2025). For $\rho_1 = 1$ with $\rho_2 > 0$ (no Mix1 with partial Mix2), the recursion grows linearly in $r$ and no uniform-in-$r$ bound exists, consistent with the consensus error being able to diverge in this regime.

*Remark* E.4 (Convexity). Assumption 1 requires $f$ to be convex, but convexity enters our argument only through the one-step contraction $\|T_\eta(x) - T_\eta(y)\|^2 \leq \|x - y\|^2$ in Theorem E.1. Extending the bound to the non-convex case (e.g., under a Polyak–Łojasiewicz condition) is left to future work.

