# OpenReview forum: "Factored Gossip DiLoCo: Reducing Blocking Communication within DiLoCo"
_ICML.cc/2026/Conference — ICML 2026 regular_

### Official Review · Reviewer_i4fh · 2026-02-22

**Soundness:** 3
**Presentation:** 3
**Significance:** 3
**Originality:** 2
**Overall Recommendation:** 5
**Confidence:** 4

**Summary:**

The paper presents `Factored Gossip DiLoCo` algorithm which aims to reduce the latency of bandwidth heavy outer synchronization step of `DiLoCo`. To achieve this, the authors factorize the averaging step of  `DiLoCo` into two steps similar to NoLoCo [1]. `Factored Gossip DiLoCo`  presents two mixes: M1 for the parameters and M2 for the local-gradients or the cumulative local updates. The paper analyzes global and local variant of these mixes where the global variant uses all_reduces where as the local variant follows the decentralized gossip mechanism by choosing disjoint random pairs in each iteration. Even with the caveat of having a noiser gradient than the classical version (i.e., the local gradients in *round* $t$ are computed on averaged parameters from *round* $t-2$), the experimental results show that GlobalM1 performs competitively to DiLoCo while drastically reducing the wall-clock training time. This is an interesting observation, however, `Factored Gossip DiLoCo` need to be validated at larger scale in terms of data and model size with all parallelisms (FSDP, TP, PP, CP, EP etc) in place to be of practical value.

1. Kolehmainen, J., Blagoev, N., Donaghy, J., Ersoy, O., and Nies, C. Noloco: No-all-reduce low communication training method for large models. arXiv preprint arXiv:2506.10911, 2025.

**Compliance With Llm Reviewing Policy:**

Affirmed.

**Final Justification:**

Authors have addressed all my concerns and therefore I retain my positive score.

**Key Questions For Authors:**

1. Experimental section mentions that `LocalM1M2` baseline is very similar to `NoLoCo` (Kolehmainen et al., 2025), though with dampening and pipeline randomization removed". It might be useful to present the exact `NoLoCo` baseline in the results as it is a strong competitor and the most similar work to  `Factored Gossip DiLoCo`. Is there a reason why exact `NoLoCo` baseline was not included in the experimental results?
2. What is the memory overhead of `Factored Gossip DiLoCo`? It is not clear from the paper if having two mixes in parallel requires additional buffers and this could result in a memory overhead proportional to model size.
3. The paper focuses on establishing non-blocking communication to enable compute-communication overlap. Therefore comparing with methods such as streaming `DiLoCo` is valuable.
4. From Figure 3, `GlobalM1` seems to be a strong competitor to `DiLoCo` and it is important to discuss the inclusion of optimizer states in the mixing to the main paper. For Table. 2, can you add `DiLoCo` results i.e., outer optimizer synchronization with optimizer states to the results? It is not clear if `GlobalM1S` still outperforms `DiLoCo` when optimizer states are synchronized.
5. In Table 1, what are the inner and outer optimizers used?
6. If possible, can you a table/discussion explaining under what regimes each of method (`DiLoCo, GlobalM1, GlobalM1LocalM2` etc) are preferred?
7. It is surprising to see that even at $1k$ local steps, `GlobalM1` performs competitive to `DiLoCo` given the lack of centralized synchronization! However, with `GlobalM1`, there is no reason to increase H more than what is needed to overlap the latency of communication step. What is the minimum $H$ value needed to completely overlap the communication of `GlobalM1` in your setup?

Minor Changes:
1. Please add a detailed pseudo-code of the proposed method in the appendix for the ease of understanding.

**Limitations:**

1. The authors should clearly discuss the memory overheads of the proposed method.

**Strengths And Weaknesses:**

Strengths:
1. The paper explores an important problem of communication overhead in large-scale distributed training.
2. The paper proposes `Factored Gossip DiLoCo` an extension and ensemble of `DiLoCo`  and `NoLoCo` to effectively handle the communication overhead of the outer optimizer of in distributed FedOpt based training setups.
3. The authors present experimental evidence and ablation studies to demonstrate the performance of various mixing strategies of the proposed  `Factored Gossip DiLoCo` algorithm. In particular, they show that `GlobalM1` performs competitively to `DiLoCo` while significantly reducing the training time.
4. The paper is well-motivated, well-written and easy to follow.

Weaknesses:
1. Proper comparison with competing baselines such as `NoLoCo` and `Streaming DiLoCo` can strengthen the claims of the paper.

---

> ### Author Rebuttal · Authors · 2026-03-31
>
> _Comparison to NoLoCo_
> --
> See response to Reviewer irQ2.
>
> _Comparison to Streaming DiLoCo_
> --
> We first note that Streaming DiLoCo's paper advocates for a maximum overlap of 5 steps, as performance degrades with more overlap (their Fig.8,10). This is enough for large compute utilization (CU) increases compared to DiLoCo in their high-bandwidth setting: 85%->95% @ 10Gbps, ->100% with quantization. However, it provides negligible gains over DiLoCo in our low bandwidth setting: 36% -> 36% CU @ 100Mbps and 53%->54% @ 200Mbps, compared to much larger gains from our methods.
>
> Streaming DiLoCo has three components: (1) parameter partitioning with different outer step schedules (reduces peak bandwidth and average drift), (2) computation-communication overlap (increases CU at the cost of staleness), and (3) quantization (reduces communication volume). (1) and (3) are orthogonal to our approach and can be incorporated with our method.
>
> **Computation-communication overlap comparison:**
> In our paper we focused on methods without staleness (lines 84–90, col. 2), hence no initial comparison. However, since we trade off noisier (but temporally correct) outer gradients in order to overlap without staleness, we now directly compare to their overlapping approach. We communicated with the authors about implementation details due to minor issues in the paper's pseudocode (and no released code).
>
> Using their recommended 5-step overlap, our reimplementation shows large val loss spikes during merging and diverges after 900 steps. To match their HPs we set $H=30$ and outer LR to 0.4, which improves stability (much smaller spikes) but still diverges after 500 steps (many more outer rounds).
>
> This was a great example of JS vs L2 dist, peak L2 dist was tiny (22.4) but JS dist skyrocketed to 3, showing that worker drift is controlled but merging stale parameters is very unstable. It also suggests that their partitioning is critical for robustness to staleness.
>
> Thus our method is much more stable, as it can even converge with the original DiLoCo HPs (high outer LR which speeds convergence), and can provide CU gains in low bandwidth settings.
>
> _Memory Overhead_
> --
> Our method does not do the two mixing operations in parallel, they are applied sequentially. There is a need for an extra copy of the model since we split $x_r$ into $x_{m,r}$ and $z_{m,r}$, which are needed at the same time. However we keep this and other outer buffers on CPUs. We discuss this in more detail (including memory overhead) in the response to Reviewer sYyU.
>
> _In Table 1, what are the inner and outer optimizers used?_
> --
> We use the same optimizers as DiLoCo: AdamW for inner, SGD with Nesterov momentum for outer. We will clarify this in the paper. We use the same hyperparameters as DiLoCo for those optimizers as well, except for a slightly higher peak inner learning rate (3e-4 vs. 4e-4).
>
> _Under what regime is each method preferred?_
> --
> The preferred method depends on the trade-off between validation perplexity and compute utilization, which is governed by $H$ and the bandwidth. For $H=100$, DiLoCo is preferable for >2Gbps, GlobalM1LocalM2 is preferable for 500Mbps - 2 Gbps, and GlobalM1 is preferable for <100Mbps (supported by Tab. 1 and Fig. 4).
>
> More generally, validation perplexity vs. time curves (e.g., Figure 3, middle and right) provide a clearer way to select the best method for a given setting ($H$ and bandwidth), but is expensive to do for many settings.
>
> _What is the minimum H value for complete communication overlap for GlobalM1_
> --
> This depends on bandwidth. In our setup, full overlap occurs at approximately $H=180$ for 100Mbps and $H=90$ for 200Mbps.
>
> _Add DiLoCo results to Table 2 for ease of comparison, add detailed pseudo-code of the proposed method in the appendix_
> --
> Good point, we will do both of these.
>
> _Other_
> --
> Just to clear up a misunderstanding in the summary: "having a noisier gradient than the classical version (i.e., the local gradients in round $t$ are computed on averaged parameters from round $t-2$)" would correspond to stale outer gradients, since it uses parameters from round $t-2$.
>
> The increased noise in our method arises because the local gradients in round $t$ are computed on locally averaged (rather than globally averaged) but temporally correct parameters from round $t-1$, but there is no staleness.

---

> > ### Author Rebuttal · Reviewer_i4fh · 2026-04-01
> >
> > The authors have responded to all my questions. I will retain my positive score for this work.

---

### Official Review · Reviewer_koP2 · 2026-03-11

**Soundness:** 2
**Presentation:** 2
**Significance:** 3
**Originality:** 3
**Overall Recommendation:** 4
**Confidence:** 3

**Summary:**

This paper decomposes DiLoCo's outer synchronization into non-blocking Mix1 and blocking Mix2 to improve compute utilization limited by blocking communication with preserving perplexity to DiLoCo. Specifically, it mixes the latest  local outer updates parameters using Mix1 but does not wait for the globally mixed value. Instead, it immediately computes the next round's local gradients using the latest local outer update to overlap communication and computation. To further reduce the communication volume of Mix2, the paper introduces a functional consensus metric based on the Jensen–Shannon (JS) distance between worker logits on a fixed batch and the logits of the averaged model. Then, by offline profiling to choose a subset of parameters for Mix2 communication, ones contribute most to the increase in JS distance when not averaged are chosen to selectively update. As a result, the proposed design improves compute utilization while achieving perplexity comparable to DiLoCo in low bandwidth settings.

**Compliance With Llm Reviewing Policy:**

Affirmed.

**Final Justification:**

I appreciate the authors for addressing my questions. As the major concerns from my review have been addressed, I raise my score to weak accept.

**Key Questions For Authors:**

1. The paper shows improved compute utilization, but it is unclear how this translates into end to end training benefits. Can the authors provide results that more directly demonstrate improved training efficiency, such as wall clock time to reach a target perplexity or overall training throughput?
2. The evaluation is limited to Llama family models. Could the authors discuss whether the proposed method generalizes to other model architectures, and, if possible, provide results showing training efficiency and validation perplexity on additional models?
3. Could the authors describe the experimental setup in more detail? In particular, is the network bandwidth identical across all workers, or are heterogeneous bandwidth settings considered? If the current setup assumes homogeneous bandwidth, could the authors provide results under more diverse network conditions?
4. In \S 5.2, the paper states that under simulated failures, all reduce must restart while mixing can continue with weaker mixing, which leads to lower utilization for DiLoCo. Could the authors provide more quantitative evidence for this claim, such as the restart cost and the corresponding gains of the proposed technique under the same failure setting?

**Limitations:**

Yes

**Strengths And Weaknesses:**

- Soundness
  - The paper does not fully demonstrate what is concretely improved compared to DiLoCo. While it reports compute utilization and perplexity relative to prior methods, these metrics are difficult to tanslate into clear end to end gains such as reduced wall-clock training time or improved overall training efficiency.
  - Evaluation is limited to Llama-family models, and the bandwidth setting is not described in detail. It is unclear whether all workers use the identical network bandwidth or whether heterogeneous network conditions were evaluated. So, the practical benefit and generality of the proposed method remain vague.

- Presentation
  - The paper is generally readable, but the presentation could be clearer in clarifying the specific improvements made compared to DiLoCo.
  - The paper lacks a sufficient justification for how the two phased design enhances efficiency over the baseline.
  - The discussion of the experimental results needs to be more precise and clear, with more explicit references to the relevant figures and tables.

- Significance
  - The paper addresses an important problem in low bandwidth distributed training for large models. The proposed ideas could be useful for future work if the paper more clearly demonstrate their practicality.

- Originality
  - The paper is moderately original.
  - It does not introduce a totally new optimization framework, but it provides a reformulation of DiLoCo synchronization and a novel use of functional consensus metrics to reduce synchronization overhead.

---

> ### Author Rebuttal · Authors · 2026-03-31
>
> _End to end training benefits are unclear ... Can the authors provide results ... such as wall clock time to reach a target perplexity_
> --
> We show this in the right two plots of Figure 3, where we show validation perplexity as a function of wall-clock time. In these plots, our methods achieve significantly better perplexity–time trade-offs than DiLoCo, demonstrating faster progress toward a given perplexity target.
>
> _Does not introduce a totally new optimization framework ... provides a reformulation of DiLoCo synchronization and a novel use of functional consensus metrics_
> --
> We disagree that our contribution is merely a reformulation of DiLoCo. Our approach combines DiLoCo with gossip and introduces two distinct communication phases, which are not present in DiLoCo. While NoLoCo also incorporates similar ideas, it proposes a single method rather than a general framework.
>
> A key novelty of our framework is that it explicitly exposes the trade-off between consensus distance and blocking/non-blocking communication time, which to our knowledge has not been studied in prior work. Using this framework, we explore configurations that reduce blocking communication while maintaining low consensus distance, with non-blocking communication compensating where needed.
>
> This is important because consensus distance directly impacts optimization performance, this observation is not only from our results, but has also been established in prior work (see discussion in Sec. 4.4).
>
> _Generalization to other models_
> --
> Our main approach does not rely on any Llama-specific components and is expected to generalize to other models. We combine and extend data-parallel techniques (DiLoCo and gossip) that have been used across a wide range of architectures (not just LLMs).
>
> The only architecture-specific component is the analysis in Figure 2, where we measure JS-distance across parameter groups and use it to demonstrate how JS-distance can guide subset selection.
>
> _Experimental Setup_
> --
> The bandwidth limit is assumed to be identical across all workers. This is a deliberate design choice, as avoiding staleness requires certain operations to wait for the slowest communication link. As a result, our setup also reflects heterogeneous environments where all workers have bandwidth at least above this minimum threshold.
>
> Figure 3 (middle) shows that even at a low bandwidth limit of 100 Mbps our methods achieve comparable or significantly better perplexity curves compared to DiLoCo.
>
> _All-reduce vs. Gossip_
> --
> Bandwidth-optimal all-reduce methods (e.g., ring or butterfly) cannot tolerate even a single worker failure during execution, as each stage depends on all workers being present. In practice, this causes the operation to hang [1], after which it must be aborted and restarted. For example, PyTorch’s TorchFT aborts the all-reduce and reruns it with a reduced set of workers once a failure is detected [2]. This restart incurs a full communication cost for the operation, reducing overall utilization.
>
> In contrast, pairwise gossip is inherently robust to such failures. If one or more workers drop out, unaffected pairs can still complete their averaging steps, and consensus within the remaining workers continues to improve. Avoiding the full restart cost and leads to higher effective utilization under failures.
>
> This difference is reflected in Fig. 4, where all-reduce–based methods show a noticeable drop in utilization under failures (or delayed drop if it is overlapped), while gossip-based communication is unaffected.
>
> - [1] https://docs.nvidia.com/deeplearning/nccl/user-guide/docs/troubleshooting.html
> - [2] https://github.com/meta-pytorch/torchft

---

> > ### Author Rebuttal · Reviewer_koP2 · 2026-04-03
> >
> > I appreciate the authors for addressing my questions. One remaining concern is about bandwidth.
> > I believe a homogeneous bandwidth cap does not adequately reflect heterogeneous environments (link heterogeneity affects tail latency, Mix1 overlap, and the advantage of global vs local communication).
> >
> > I believe the other points are reasonable choices given the paper's scope, but would benefit from discussion in future work.

---

> > > ### Author Response · Authors · 2026-04-07
> > >
> > > _a homogeneous bandwidth cap does not adequately reflect heterogeneous environments_: that is a good point, it depends on what type of heterogeneity is considered.
> > >
> > > If all workers have stable (low variability) communication times that differ between workers, then a homogeneous bandwidth cap is conservative but valid, as it effectively models the system at the speed of the slowest link.
> > >
> > > However, if workers have stochastic communication times with high variability (e.g., long-tailed communication times that are potentially unbounded), then a homogeneous bandwidth cap is insufficient. One way to address this (again conservative) is to set a limit and timeout (fail) communication that exceed the limit.
> > >
> > > This is similar to our communication with failures setting (line 360 col 1), where a 5% communication failure rate is investigated. In that setting, we observe that failures introduce significant slowdowns for global averaging (all-reduce) due to restart requirements, while local mixing does not have any slowdown (at the cost of weaker mixing). Thus, this favours our approach over vanilla DiLoCo: local mixing has no slowdown, and any global communication is Mix1 communication which often has more overlap time before it becomes blocking.

---

### Official Review · Reviewer_sYyU · 2026-03-12

**Soundness:** 2
**Presentation:** 3
**Significance:** 2
**Originality:** 3
**Overall Recommendation:** 4
**Confidence:** 3

**Summary:**

This paper addresses the communication bottleneck and brittleness of DiLoCo's global outer synchronization in low-bandwidth, decentralized training environments. The authors propose "Factored Gossip DiLoCo," a novel framework that relaxes exact synchronization into two distinct approximate mixing operations: a non-blocking parameter mix (Mix1) that overlaps with local computation without introducing temporal staleness, and a blocking pseudo-gradient mix (Mix2) that enforces consensus to maintain optimization stability. Furthermore, the authors leverage JS distance to measure functional disagreement between workers, discovering that applying the blocking Mix2 step to only a subset of highly sensitive parameters (such as token embeddings and the LM head) optimizes the trade-off between compute utilization and model stability. Empirical results training up to a 1.5B parameter language model demonstrate that this factored approach substantially improves compute utilization and robustness to transient network failures compared to baseline DiLoCo.

**Compliance With Llm Reviewing Policy:**

Affirmed.

**Final Justification:**

This work presents a technically sound and well-motivated approach to decentralized training by factorizing outer synchronization into distinct mixing operations. However, I recommend a weak accept rather than a higher score because the improved compute utilization comes at the direct expense of data and token efficiency. While wall-clock time is reduced, the models achieve worse validation perplexity than baseline DiLoCo or Sync-DP for the same iteration or data budget. Furthermore, the theoretical consensus error result relies on strong convexity and unbiased drift assumptions that oversimplify the actual non-convex dynamics of large-scale language model training. Although the authors' introduction of JS distance is a clever empirical tool for monitoring functional disagreement, the framework ultimately acts as a tunable trade-off between stability and speed rather than a method that preserves baseline performance. Consequently, while the paper is a useful contribution to the field, these persistent performance gaps and theoretical limitations temper its overall significance.

**Key Questions For Authors:**

Q1 (**non-convex**): Are there plans to extend this theoretical framework to non-convex settings (e.g., utilizing the Polyak-Łojasiewicz condition), and if so, how might the relationship between the contraction factors ($\rho_1$, $\rho_2$) and the consensus error change?

Q2 (**JS distance**): Since Jensen-Shannon distance proved to be a vastly superior empirical proxy for optimization stability compared to L2 distance, is it possible to formulate a theoretical bound directly based on functional/distributional divergence rather than parameter space?

Q3 (**downstream tasks**): A degradation of ~1.0 in validation perplexity can sometimes translate to significant performance drops in actual downstream capabilities (e.g., reasoning or instruction following). Have you evaluated the final GlobalM1LocalM2-subset model on zero-shot or few-shot downstream benchmarks to quantify this impact?

Q4 (**memory overhead**): Could you explicitly profile and report the peak GPU memory (VRAM) overhead required to maintain the overlapping buffers for Mix1 compared to the baseline DiLoCo implementation?

**Limitations:**

yes

**Strengths And Weaknesses:**

Strength:
1. The paper introduces a clever factorization of DiLoCo's outer synchronization into a non-blocking step (Mix1) and a blocking step (Mix2). Crucially, unlike other recent asynchronous DiLoCo variants, this overlapping of communication and computation does not introduce temporal staleness.
2. The authors successfully demonstrate that standard L2 parameter distance does not correlate perfectly with optimization instability. Instead, they use Jensen-Shannon (JS) distance on model logits to track functional disagreement, proving it to be a much better predictor of training instability.
3. The research directly targets consumer-grade, low-bandwidth decentralized settings (50Mbps to 1Gbps) rather than assuming ideal datacenter conditions. The GlobalM1LocalM2-subset method provides another option for a trade-off between compute utilization and convergence.

Weakness:
1. The mathematical bounds provided for consensus error (Equation 27) rely on strong simplifying assumptions like convexity. The authors acknowledge that these assumptions limit the direct translation of L2 distance theory to actual function loss, meaning the theory does not fully explain the complex dynamics of LLM training.
2. The proposed method successfully improves compute utilization, but this comes at the direct expense of final model quality. Sync-DP provides the best model quality but terrible utilization (1%) under low-bandwidth constraints, while Factored DiLoCo pushes utilization up to 36%–82% but degrades the model's perplexity. A major weakness is that the framework does not actually solve the communication bottleneck without sacrificing the theoretical convergence guarantees and empirical performance of exact synchronization.
3. To make Mix1 non-blocking or asynchronous, the communication must overlap with the next round's local computations. Mathematically, workers are computing inner optimization steps on their local parameters ($z_{m,r}$) while simultaneously sending/receiving data to compute the globally mixed parameters ($x_{m,r}$). This inherently implies the system must hold an additional buffer (an extra copy of the model parameters) in memory to handle the background network transfers while the GPU actively updates the local weights.

---

> ### Author Rebuttal · Authors · 2026-03-31
>
> _Are there plans to extend this theoretical framework to non-convex settings?_
> --
>
> We follow Khaled et al. (2025) for our theoretical framework, who have given the only convergence proof for DiLoCo as far as we are aware. Thus, we inherit their assumptions, including convexity, and introduce one more (needed to use the same proof strategy as them, essentially says the worker drift is unbiased).
>
> While we do not currently provide a non-convex analysis, our result's dependence on convexity is for a non-expansiveness lemma. This could be extended to non-convex settings under a suitable stability condition (perhaps PL), where a similar contraction argument could be used.
>
> _Is it possible to formulate a theoretical bound directly based on functional/distributional divergence rather than parameter space?_
> --
> We are not aware of prior work that derives convergence bounds for a distributed or decentralized optimizer directly in terms of JS-distance between worker distributions.
>
> Our understanding is that this is challenging, as it would require
> - (1) relating bounded parameter updates to bounded changes in JS-distance between worker distributions, and
> - (2) establishing a recursion analogous to standard consensus analyses: bounding distance to the optimum in terms of both the average model and the disagreement (measured in JS-distance) between workers.
>
> While (1) can be done for L2 distance through Lipschitz assumptions (what our theory based on Khaled et al. does), extending it to JS-distance would require additional steps: (i) controlling how parameter perturbations affect logits/output probabilities, and (ii) translating logit differences into KL/JS divergence bounds that are sufficiently tight for analysis. It is less clear how to formulate (2) directly in terms of JS-distance.
>
> Furthermore, JS-distance is data dependent, introducing further assumptions on the input distribution. However, in our experiments we found it to be stable when computed on a sufficiently large fixed random batch.
>
> We therefore view developing a theory directly in terms of functional/distributional divergence as an interesting and important direction for future work.
>
>
> _A degradation of ~1.0 in validation perplexity can sometimes translate to significant performance drops in actual downstream capabilities_
> --
>
> We agree that a change of ~1.0 in validation perplexity can correspond to a decrease in downstream performance. However, this difference occurs when comparing perplexity at a fixed token budget (or number of inner steps). When instead considering validation perplexity as a function of wall-clock time (Figure 3 middle and right) our methods perform substantially better than DiLoCo (with only GlobalM1LocalM2 being slightly worse for 100 Mbps).
>
> As pretraining is usually done now with 5-15T tokens, for decentralized training in low bandwidth settings _time_ is the limiting factor rather than _token efficiency_ (validation perplexity for a token budget). In our main configuration (M=8, H=100) with a 200Mbps bandwidth limit, 1T tokens would take DiLoCo approximately 29 weeks, compared to 16 weeks for GlobalM1.
>
> Additionally, we have completed the experiments missing from Table 1. These show that, across a range of configurations, validation perplexity at a fixed token budget remains close to DiLoCo, particularly for GlobalM1LocalM2:
>
> | Experiment | DDP | DiLoCo | GlobalM1LocalM2 | GlobalM1 | LocalM1M2 |
> |---|---|---|---|---|---|
> | 30B tokens | 14.85 | 15.52 | 15.60 | 15.91 | 15.83 |
> | H = 500 | - | 22.67 | 22.78 | 23.17 | 23.13 |
> | H = 1k | - | 28.17 | 28.13 | 28.65 | 28.82 |
> | M=16 (Global BS fixed) | 17.22 | 19.99 | 20.70 | 21.39 | 22.24 |
> | M=16 (Local BS fixed) | 18.17 | 20.91 | 21.48 | 22.26 | 23.01 |
>
> _Memory overhead_
> --
> In our implementation there is no difference in peak VRAM between DiLoCo and any of our methods.
>
> This is because we place the entire outer step on the CPU, including all tensors (outer parameters, outer gradients, outer optimizers state, and the extra copy described below) as well as the associated operations (all-reduce/mixing via CPU GLOO, gradient computation via subtraction, and the outer optimizer step). This design is practical, as decentralized communication already requires CPU–GPU transfers, and the communication cost dominates the relatively lightweight outer-step computations.
>
> You are right, we need one extra copy of the model for our outer step compared to DiLoCo since $x_{m,r}$ and $z_{m,r}$ must both exist during the calculation of the outer gradient, but this is also on the CPU.
>
> On each GPU (one worker in our implementation) the peak VRAM during local steps is 28.29GB, and during the outer step the peak VRAM rises to 29.89GB. This should be for buffers when copying to/from the CPU (we copy/process one parameter tensor at a time).
>
> Importantly, all reported peak VRAM values are identical between our implementation of DiLoCo and our proposed methods.

---

> > ### Author Rebuttal · Reviewer_sYyU · 2026-04-03
> >
> > I thank the authors for the thoughtful responses and the additional experiments. I think my concerns are somehow resolved.
> >
> > The only concern I have in mind is the data / compute efficiency of the proposed methods. Even though the proposed methods can achieve better per-iteration runtime, they worsen the losses with the same iteration / data budget. The loss trends seem to be flat as the training goes on, so it may take many more iterations for the proposed methods to achieve the same level of validation loss as Sync-DP.
> >
> > The work is solid and interesting in general. Therefore, I will keep my positive score.

---

### Official Review · Reviewer_irQ2 · 2026-03-13

**Soundness:** 3
**Presentation:** 3
**Significance:** 3
**Originality:** 2
**Overall Recommendation:** 5
**Confidence:** 4

**Summary:**

Distributed training of large models outside data centers is an open problem, as it leads to high communication overheads during synchronization. DiLoCo partially addresses this problem by reducing the global synchronization steps across workers. However, the all-reduce blocking communication in DiLoCo still hampers compute utilization and leads to slower training. The authors propose Factored Gossip DiLoCo, where they factor out the communication into (1) an approximate global communication Mix1 that removes temporal staleness, and (2) an exact blocking gossip communication to reduce disagreement. While this is an approximate method that introduces noise in the training, the authors provide a theoretical analysis of the method and experimental evidence that it improves compute utilization while having a limited effect on the model performance.

**Compliance With Llm Reviewing Policy:**

Affirmed.

**Final Justification:**

Overall, this is a strong paper. The rebuttal was convincing, and the authors promised to make the required edits. I recommend accepting it.

**Key Questions For Authors:**

Please address the weaknesses above.

**Limitations:**

Yes, addressed in text.

**Strengths And Weaknesses:**

## Strengths

[S1] This is an important and timely problem given the ever-growing size of models and the energy demands of training models within a data center.

[S2] Theoretical formulation and grounding are present.

[S3] The paper is well-written and easy to follow.

[S4] The components are described, motivated, and evaluated well. I really like the insights from JS distance and the train loss instability presented in Figure 7.

[S5] The experiments justify the claims with different numbers of workers, 2 different low-end bandwidths, and 3 model parameter sizes. Furthermore, baselines comprisign variants of local and global communication provide strong insights useful for future research in the domain.

Overall, I am pretty positive about the paper. Some weaknesses listed below can be addressed with some additional experiments or clarifications.

## Weaknesses

[W1] There is a mismatch in limiting the scope to *exact communication* in line 84 (col. 2), where the authors mention that sparsification methods are out of scope, and then introducing subset local M2, where only a subset of parameters are exchanged. The local M2 subset is a special form of sparsification that brings other sparsification methods within the scope and should be evaluated against.

[W2] While LocalM2-subset is motivated using JS distance, an ablation of the impact on training for various lengths of the subset is not present. This experiment would definitely make the paper stronger.

[W3] In the experimental setup, the limits of the system are not well tested. In particular, it is unclear what happens if we reduce the bandwidth further. Furthermore, the simulated failure is limited to only 5% of the communication, which is quite low for the low number of nodes in the system.

[W4] The authors claim that the method matches DiLoCo's training progress (in abstract, and *comparable* in intro). It is important to rephrase this and be honest about not matching the final performance of DiLoCo.

[W5] The work is quite close to NoLoCo as the authors acknowledged. However, I wonder why the full version of NoLoCo is not evaluated against.

## Nits
Line 244 (col. 2) *averaging* typo.

---

> ### Author Rebuttal · Authors · 2026-03-31
>
> _Comparison to NoLoCo_
> --
> As we mentioned in our paper, LocalM1M2 is essentially NoLoCo with dampening and pipeline randomization removed.
>
> **Why we do not compare with pipeline randomization**: Our paper focuses on efficient optimization within bandwidth-constrained settings along the **data-parallel (DP) axis**, so we isolate effects from that axis. NoLoCo's pipeline randomness operates along the pipeline-parallel (PP) axis and requires architectures that support PP with random routing, whereas our method applies to any model as a pure DP approach.
>
> Pipeline randomness could easily be incorporated into our method and may further improve our results, but this is outside our scope (we will add this to the future work section). In NoLoCo, pipeline randomness is critical for performance and stability (shown in their paper and discussed more below), while our approach achieves stable and efficient optimization using only DP techniques.
>
> **Comparison with dampening**: We did not initially test this as it appears to be a weaker version of our LocalM1M2. We now compare to NoLoCo without PP, so using their exact outer optimizer step (local Mix1 but weakly as a dampening term).
>
> We implemented and ran this on a 1.5B model using our hyperparameters (HP): DiLoCo's outer HP (outer LR 0.7, outer mom 0.9), H=100 and inner LR 3e-4 (DiLoCo experiments on multiple H and uses inner LR 4e-4 but on much smaller models); and NoLoCo's dampening HP (tied to the outer LR as per their code). This diverged after 5k steps. We then used their paper's HP for their 1.3B param model: outer mom 0.5, H=50, inner LR 2e-4. This converged, but performed much worse: perplexity of 22.65 at 10B tokens vs 18.65-19.42 for DiLoCo and our methods (including LocalM1M2).
>
> This was surprising as H=50 should outperform H=100 (at 2x communication cost vs H=100). This suggests pipeline randomness is important for both their performance and stability, though their method still appears unstable.
>
> Inspecting their formulation, their outer step does not use true Nesterov momentum (no lookahead), despite being described as a "modified Nesterov momentum" approach, which may explain the gap. Prior work shows Nesterov is critical for the performance of pseudo-gradient methods: first shown in DiLoCo's paper and then more thoroughly in SNOO [1]. We verified that their official code matches their paper.
>
> - [1] Kallusky, Dominik & Rao, Vinay & Nandavanam, Vishal & Shi, Hao-Jun. (2025). SNOO: Step-K Nesterov Outer Optimizer - The Surprising Effectiveness of Nesterov Momentum Applied to Pseudo-Gradients. 10.48550/arXiv.2510.15830.
>
> _Mismatch in limiting the scope to exact communication_
> --
> That is a good point. We limited the scope because quantization and sparsification are orthogonal directions that can easily be applied on top of our method. The subset approach falls under this, but the intention was to demonstrate our observation that JS distance is a really good signal. We then decided it was also a good way to show the flexibility of the general approach (Mix2 can be very lightweight), conflating this initial intention. We will make the original intention more clear in the paper.
>
> There is definitely more room for exploration on how lightweight Mix2 can get while significantly improving stability over GlobalM1 using sparsification and quantization strategies.
>
> _Subset ablations_
> --
> Since the intention was to control JS distance, we ablated peak JS distance to determine that subset (meaning we only needed to run experiments until just after the LR peak).
>
> | Subset | Peak JS-dist |
> |---|---|
> | Token Emb + LM-Head  | 0.28 |
> | + Layer 0  | 0.26 |
> | + Layer 1  | 0.22 |
>
> _Limits of the system_
> --
> As bandwidth drops further all methods converge to the same very low compute utilization (see Figure 4 top). We chose 5% failures as a conservative estimate for what standard large scale pre-training (DiLoCo's setup) would be like (looking at Meta's fault tolerant writeups). Higher failure rates (likely in decentralized setups) would definitely favor our system more, and we actually envision changing the Mix1 global average from ring all-reduce to gossip based (push-sum algorithm). This would be much more fault tolerant, though a lot slower, which may be fine since Mix1 is non-blocking.
>
> _Matching claim_
> --
> We will rephrase this to be softer. Matching is not correct and will be changed, however GlobalM1LocalM2 is comparable to DiLoCo (18.80 vs 18.65 at 10B tokens, 15.60 vs 15.52 at 30B tokens). This can also be seen in the validation perplexity curve in Figure 3 left: the gap between the two decreases quickly. The other proposed methods lag slightly behind DiLoCo as shown in that plot (we will make this clear). We expanded on our experiments in the response to Reviewer sYyU, which show that GlobalM1LocalM2 is also comparable to DiLoCo in a range of settings, and performs slightly better than DiLoCo for H=1k steps.

---

> > ### Author Rebuttal · Reviewer_irQ2 · 2026-04-03
> >
> > Thank you for the detailed rebuttal. I continue to be positive about this work and will maintain my score as *accept*.

---

### Decision · Program_Chairs · 2026-04-30

**Decision:**

Accept (regular)

**Comment:**

This paper introduces a novel approach to decentralized training at internet scale, specifically designed for public settings. The core idea is to decouple communication into two distinct components: an approximate global communication, which eliminates temporal staleness by ensuring timely updates, and an exact blocking gossip communication mechanism, which reduces disagreement among nodes by synchronizing model parameters precisely.

The proposed method addresses a critical challenge in decentralized training: the tension between communication efficiency and model consistency. Unlike prior asynchronous variants of DiLoCo, this approach does not introduce temporal staleness, often due to overlapping communication and computation, which maintains accuracy and efficiency. In addition, they introduce a novel way to use Jensen-Shannon  distance on model logits to track functional disagreement, demonstrating that this distance is a superior predictor of training instability compared to traditional metrics, e.g., L^2 norm.

While reviewers raised valid concerns, such as the proximity of this approach to NoLoco and the need for stronger mathematical correspondence between the algorithm and their proofs, the work remains fundamentally sound. Empirical results confirm that this method outperforms relevant baselines, delivering practical improvements in both stability and convergence. The novel integration of these techniques, particularly the absence of temporal staleness, represents a significant advancement in the field. For these reasons, I recommend the acceptance of this work.